# PREDICTING PHYSICS IN MESH-REDUCED SPACE WITH TEMPORAL ATTENTION

**Xu Han** *
Tufts University
Xu.Han@tufts.edu

**Han Gao**\*
University of Notre Dame
hgao1@nd.edu

**Tobias Pffaf**\*
DeepMind
tob.pfaff@gmail.com

**Jian-Xun Wang**
University of Notre Dame
jwang33@nd.edu

**Li-Ping Liu**
Tufts University
Liping.Liu@tufts.edu

## ABSTRACT

Graph-based next-step prediction models have recently been very successful in modeling complex high-dimensional physical systems on irregular meshes. However, due to their short temporal attention span, these models suffer from error accumulation and drift. In this paper, we propose a new method that captures long-term dependencies through a transformer-style temporal attention model. We introduce an encoder-decoder structure to summarize features and create a compact mesh representation of the system state, to allow the temporal model to operate on a low-dimensional mesh representations in a memory efficient manner. Our method outperforms a competitive GNN baseline on several complex fluid dynamics prediction tasks, from sonic shocks to vascular flow. We demonstrate stable rollouts without the need for training noise and show perfectly phase-stable predictions even for very long sequences. More broadly, we believe our approach paves the way to bringing the benefits of attention-based sequence models to solving high-dimensional complex physics tasks.

## 1 INTRODUCTION

There has been an increasing interest in many scientific disciplines, from computational fluid dynamics [3, 40] over graphics [43, 41] to quantum mechanics [21, 1], to accelerate numerical simulation using learned models. In particular, methods based on Graph Neural Networks (GNN) have shown to be powerful and flexible. These methods can directly work with unstructured simulation meshes, simulate systems with complex domain boundaries, and adaptively allocate computation to the spatial regions where it is needed [8, 39, 35, 52].

Most models for complex physics prediction tasks, in particular those on unstructured meshes, are next-step prediction models; that is, they predict the next state $u(t+1, x)$ of a physical system from the current state $u(t, x)$. As next-step models suffer from error accumulation, mitigation strategies such as training noise [39] or augmented training data using a solver-in-the-loop [42] have to be used to keep rollouts stable. These remedies are not without drawbacks– training noise can be hard to tune, and ultimately place a bound on the achievable model accuracy. And worse, next-step models also tend to show drift, which is not as easily mitigated. Failure examples include failure to conserve volume or energy, shift in phase, or loss of shape information (see e.g., the failure case example in [39]).

On the other hand, auto-regressive sequence models such as Recurrent neural networks (RNNs), or more recently transformers, have been hugely successful in predicting sequences in NLP and image applications [36, 31]. They can capture stochastic dynamics and work with partial observations [50]. Furthermore, their long attention span allows them to better preserve phase and conserved quantities [18]. However, as memory cost for full-sequence transformer models scales with both

---

*Equal contribution.

sequence length and spatial extents, it is hard to directly extend such models to predict physical systems defined on large unstructured meshes.

This paper combines powerful GNNs and a transformer to model high-dimensional physical systems on meshes. The key idea is creating a locally coarsened yet more expressive graph, to limit memory consumption of the sequence model, and allow effective training. We first use a GNN to aggregate local information of solution fields of a dynamic system into node representations by performing several rounds of message passing, and then coarsen the output graph to a small set of pivotal nodes. The pivotal nodes' representations form a latent that encodes the system state in a low-dimensional space. We apply a transformer model on this latent, with attention over the whole sequence, and predict the latent for the next step. We then use a second GNN to recover the full-sized graph by up-sampling and message passing. This procedure of solving on a coarse scale, upsampling, and performing updates on a fine-scale is related to a V-cycle in multigrid methods [4].

We show that this approach can outperform the state-of-the-art MeshGraphNets[35] baseline on accuracy over a set of challenging fluid dynamics tasks. We obtain stable rollouts without the need to inject training noise, and unlike the baseline, do not observe strong error accumulation or drift in the phase of vortex shedding.

## 2 RELATED WORK

Developing and running simulations of complex, high-dimensional systems can be very time-intensive. Particular for computational fluid dynamics, there is considerable interest in using neural networks for accelerating simulations of aerodynamics [3] or turbulent flows [22, 46]. Fast predictions and differentiable learned models can be useful for tasks from airfoil design [40] over weather prediction [37] to visualizations for graphics [43, 41, 7]. While many of these methods use convolutional networks on 2D grids, recently GNN-based learned simulators, in particular, have shown to be a very flexible approach, which can model a wide range of systems, from articulated dynamics [38] to dynamics of particle systems [24, 39]. In particular, GNNs naturally enable simulating on meshes with irregular or adaptive resolution [35, 8].

These methods are either steady-state or next-step prediction models. There are, however, strong benefits of training on the whole predicted sequence: next-step models tend to drift and accumulate error, while sequence models can use their long temporal range to detect drift and propagate gradients through time to prevent error accumulation. Models based on RNNs have been successfully applied to 1D time series and small n-body systems [6, 54] or small 2D systems [50]. More recently, transformer models [44], which have been hugely successful for NLP tasks [36], are being applied to low-dimensional physics prediction tasks [18]. However, applying sequence models to predict high-dimensional systems remains a challenge due to their high memory overhead. Dimensionality reduction techniques, such as CNN autoencoders [34, 33, 27, 23, 30, 17, 12, 28], POD [45, 49, 5, 32, 19, 9, 48, 11], or Koopman operators [25, 10, 15] can be used to construct a low-dimensional latent space. The auto-regressive sequence model then operates on these linear (POD modes) or nonlinear (CNNs) latents. However, these methods cannot directly handle unstructured data with moving or varying-size meshes, and many of them do not consider parameter variations. For example, POD cannot operate on state vectors with different lengths (e.g., variable mesh sizes data in Fig.2 bottom). On the other hand, CNN auto-encoders can only be applied to rectangular domains and uniform grids; and rasterization of complex simulation domains to uniform grids are known to be inefficient and are linked to many drawbacks as discussed in [13].

In contrast, our method reduces the input space locally by aggregating information on a coarsened graph. This plays to the strength of GNNs to learn local universal rules and is very closely related to multi-level methods [51, 16] and graph coarsening [14, 53, 2].

## 3 METHODOLOGY

### 3.1 PROBLEM DEFINITION AND OVERVIEW

We are interested in predicting spatiotemporal dynamics of complex physical systems (e.g., fluid dynamics), usually governed by a set of nonlinear, coupled, parameterized partial differential equations

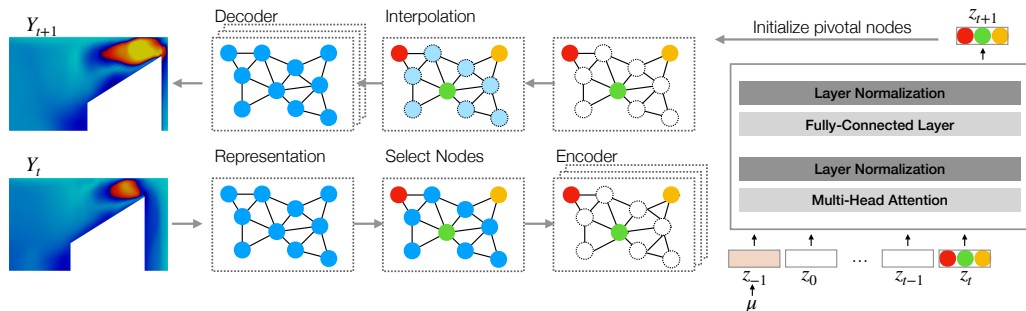

Figure 1: The diagram of the proposed model, GMR-Transformer-GMUS. We first represent the domain as a graph and then select pivotal nodes (red/green/yellow) to encode information over the entire graph. The encoder GMR runs *Message passing* along graph edges so that the pivotal nodes collect information from nearby nodes. The latent vector $z_t$ summarizes information at the pivotal nodes, and represents the whole domain at the current step. The transformer will predict $z_{t+1}$ based on *all* previous state latent vectors. Finally, we decode $z_{t+1}$ through message passing to obtain the next-step prediction $\mathbf{Y}_{t+1}$.

(PDEs) as shown in its general form,

$$\frac{\partial \boldsymbol{u}(\boldsymbol{x}, t)}{\partial t} = \mathcal{F}\big[\boldsymbol{u}(\boldsymbol{x}, t); \boldsymbol{\mu}\big], \quad \boldsymbol{x}, t \in \Omega \times [0, T_e], \tag{1}$$

where $\boldsymbol{u}(\boldsymbol{x}, t) \in \mathbb{R}^d$ represents the state variables (e.g., velocity, pressure, or temperature) and $\mathcal{F}$ is a partial differential operator parameterized by $\boldsymbol{\mu}$. Given initial and boundary conditions, unique spatiotemporal solutions of the system can be obtained. One popular method of solving these PDE systems is the finite volume method (FVM) [29], which discretizes the simulation domain $\Omega$ into an unstructured mesh consisting of $N$ cells $(C_i : i = 1, \ldots, N)$. At time $t$, the discrete state field $\mathbf{Y}_t = \{\boldsymbol{u}_{i,t} : i = 1, \ldots, N\}$ can thus be defined by the state value $\boldsymbol{u}_{i,t}$ at each cell center. Traditionally, solving this discretized system involves sophisticated numerical integration over time and space. This process is often computationally expensive, making it infeasible for real-time predictions or applications requiring multiple model queries, e.g., optimization and control. Therefore, our goal is to learn a simulator, which, given the initial state $\mathbf{Y}_0$ and system parameters $\boldsymbol{\mu}$, can rapidly produce a rollout trajectory of states $\mathbf{Y}_1 ... \mathbf{Y}_T$.

As mentioned above, solving these systems with high spatial resolution by traditional numerical solvers can be quite expensive. In particular, propagating local updates over the fine grid, such as pressure updates in incompressible flow, can require many solver iterations. Commonly used technique to improve the efficiency include *multigrid methods* [47], which perform local updates on both the fine grid, as well as one or multiple coarsened grids with fewer nodes, to accelerate the propagation of information. One building block of multigrid methods is the *V-cycle*, which consists of down-sampling the fine to a coarser grid, performing a solver update, up-sampling back on the fine grid, and performing an update on the fine grid.

Noting that GNNs excel at local updates while the attention mechanism over temporal sequences allows long-term dependency (see remark A.2), we devise an algorithm inspired by the multigrid *V-cycle*. For each time step, we use a GNN to locally summarize neighborhood information on our fine simulation mesh into pivotal nodes, which form a coarse mesh (section 3.2). We use temporal attention over the entire sequence in this lower-dimensional space (section 3.3), upsample back onto the fine simulation, and perform local updates using a mesh recovery GNN (section 3.2). This combination allows us to efficiently make use of long-term temporal attention for stable, high-fidelity dynamics predictions.

## 3.2 GRAPH REPRESENTATION AND MESH REDUCTION

**Graph Representation.** We construct a graph $\mathcal{G} = (\mathcal{V}, \mathcal{E})$ to represent a snapshot of a dynamic system at time step $t$. Here each node $i \in \mathcal{V}$ corresponds the mesh cell $C_i$, so the graph size is $|\mathcal{V}| = N$. The set of edges $\mathcal{E}$ are derived from neighboring relations of cells: if two cells $C_i$ and $C_j$

are neighbors, then two directional edges $(i, j)$ and $(j, i)$ are both in $\mathcal{E}$. We fix this graph for all time steps. At each step $t$, each node $i \in V$ uses the local state vector $\boldsymbol{u}_{i,t}$ as its attribute. Therefore, $(\mathcal{G}, (\boldsymbol{Y}_0, \ldots, \boldsymbol{Y}_T))$ forms a temporal graph sequence. The goal of the learning model is to predict $(\boldsymbol{Y}_1, \ldots, \boldsymbol{Y}_T)$ given $\mathcal{G}$ and $\boldsymbol{Y}_0$.

**Mesh Reduction.** After representing the system as a graph, we use a GNN to summarize and extract a low-dimensional representation $\boldsymbol{z}_t$ from $\boldsymbol{Y}_t$ for each step $t$. In this part of discussion, we omit the subscript $t$ for notational simplicity. We refer to the encoder as Graph Mesh Reducer (GMR) since its role is coarsening the mesh graph. GMR first selects a small set $\mathcal{S} \subseteq \mathcal{V}$ of pivotal graph nodes and locally encodes the information of the entire graph into representations at these nodes. By operating on rich, summarized node representations, the dynamics of the entire system is well-approximated even on this coarser graph.

There are a few considerations for selection of pivotal nodes, for example, the spatial spread and the centrality of nodes in the selection. We generally use uniform sampling to select $\mathcal{S}$ from $\mathcal{V}$. This effectively preserves the density of graph nodes over the simulation domain, i.e. pivotal nodes are more concentrated in important regions of the simulation domain. More details and visualizations on the node selection process can be found in section A.6.

GMR is implemented as a Encode-Process-Decode (EPD) GraphNet [39]. GMR first extracts node features and edge features from the system state using the node and edge Multi-Layer Perceptrons (MLPs).

$$\boldsymbol{v}_i^0 = \mathrm{mlp}_v(\boldsymbol{Y}[i]), \quad \boldsymbol{e}_{ij}^0 = \mathrm{mlp}_e(p(i) - p(j)). \tag{2}$$

Here $\boldsymbol{Y}[i]$ is the $i$-th row of $\boldsymbol{Y}$, i.e. the state vector at each node, and $p(i)$ is the spatial position of cell $C_i$.

Then GMR uses $L$ GraphNet processer blocks [38] to further refine node representations through message passing. In this process a node can receive and aggregate information from all neighboring nodes within graph distance $L$. Each processor updates the node and edge representations as

$$\boldsymbol{e}_{ij}^\ell = \mathrm{mlp}_\ell^e \left( \boldsymbol{e}_{ij}^{\ell-1}, \boldsymbol{v}_i^{\ell-1}, \boldsymbol{v}_j^{\ell-1} \right), \quad \boldsymbol{v}_i^\ell = \mathrm{mlp}_\ell^v \left( \boldsymbol{v}_i^{\ell-1}, \sum_{j \in \mathcal{N}_i} \boldsymbol{e}_{ij}^{\ell-1} \right), \quad \ell = 1, \ldots, L. \tag{3}$$

Here $\boldsymbol{v}_i^0, \boldsymbol{e}_{ij}^0$ are the outputs of Equation 2, and $\mathcal{N}_i$ denotes all neighbors of node $i$. The two functions $\mathrm{mlp}_\ell^e(\cdot)$ and $\mathrm{mlp}_\ell^v(\cdot)$ respectively concatenate their arguments as vectors and then apply MLPs. The calculation in equation 2 and equation 3 computes a new set of node representations $\boldsymbol{V} = (\boldsymbol{v}_i^L : i \in \mathcal{V})$ for the graph $\mathcal{G}$.

Finally GMR applies an MLP to representations of the pivotal nodes in $\mathcal{S}$ *only* to "summarize" the entire graph onto a coarse graph:

$$\boldsymbol{h}_i = \mathrm{mlp}_r(\boldsymbol{v}_i^L), \quad i \in S \tag{4}$$

We concatenate these vectors into a single latent $\boldsymbol{z} = \mathrm{concat}(\boldsymbol{h}_i : i \in S)$ as reduced vector representation of the entire graph. We collectively denote these three computation steps as $\boldsymbol{z} = \mathrm{GMR}(\mathcal{G}, \boldsymbol{Y})$. The latents $\boldsymbol{z}$ can be computed independently for each time step $t$ and will be used as the representation for the attention-based simulator.

**Mesh Recovery** To recover the system state from the coarse vector representation $\boldsymbol{z}$, we define a Graph Mesh Up-Sampling (GMUS) network. The key step of GMUS is to restore information on the full graph from representations of pivotal nodes. This procedure is the inverse of operation of GMR. We first set the representations at pivotal nodes by splitting $\boldsymbol{z}$, that is, $\boldsymbol{r}_i = \boldsymbol{h}_i, i \in \mathcal{S}$. We then compute representations of non-pivotal nodes by spatial interpolation [2]: for a non-pivotal node $j$, we choose a set $\mathcal{N}_j'$ of $k$ nearest pivotal nodes in terms of spatial distance and compute the its representations $\boldsymbol{r}_j$ by

$$\boldsymbol{r}_j = \sum_{i \in \mathcal{N}_j'} \frac{w_{ij} \boldsymbol{h}_i}{\sum_{i \in \mathcal{N}_j'} w_{ij}}, \quad w_{ij} = \frac{1}{d(j, i)^2} \tag{5}$$

Here $d(j, i)$ is the spatial distance between cells $C_j$ and $C_i$. Then every node $i \in \mathcal{V}$ has a representation $\boldsymbol{r}_i$, and all nodes' representations are collectively denoted as $\boldsymbol{R} = (\boldsymbol{r}_i : i \in \mathcal{V})$.

Similar to GMR, GMUS applies EPD GraphNet to the initial node representation $\boldsymbol{R}$ to restore $\boldsymbol{Y}$ on the full graph $\mathcal{G}$. We denote the chain of operations so far as $\hat{\boldsymbol{Y}} = \text{GMUS}(\mathcal{G}, \boldsymbol{z})$. Details about information flows in GMR and GMUS can be found in section A.1.

We train GMR and GMUS as an auto-encoder over all time steps and sequences. For each time step $t$, we compute $\hat{\boldsymbol{Y}}_t = \text{GMUS}(\mathcal{G}, \text{GMR}(\mathcal{G}, \boldsymbol{Y}_t))$ and minimize the reconstruction loss

$$\mathcal{L}_{graph} = \sum_{n=1}^{T} \|\boldsymbol{Y}_t - \hat{\boldsymbol{Y}}_t\|_2^2 \quad . \tag{6}$$

With these two modules we can encode system states $(\boldsymbol{Y}_1, \ldots, \boldsymbol{Y}_T)$ to latents $(\boldsymbol{z}_1, \ldots, \boldsymbol{z}_T)$ as a low-dimensional representation of system dynamics. In the next section, we train a transformer model to predict the sequences of latents.

### 3.3 ATTENTION-BASED SIMULATOR

As a dynamics model, we learn a simulator which can predict the sequence $(\boldsymbol{z}_1, \ldots, \boldsymbol{z}_T)$ autoregressively based on an initial state $\boldsymbol{z}_0 = \text{GMR}(\mathcal{G}, \boldsymbol{Y}_0)$ and the system parameters $\boldsymbol{\mu}$. The system parameters include conditions such as different Reynolds numbers, initial temperatures, or/and parameters of the spatial domain (see Table 3 for details). The model is conditioned on $\boldsymbol{\mu}$, to be able to predict flows with arbitrary system parameters at test time. We encode $\boldsymbol{\mu}$ into a special parameter token $\boldsymbol{z}_{-1}$ of the same length as the state representation vectors $\boldsymbol{z}_t$

$$\boldsymbol{z}_{-1} = \text{mlp}_p(\boldsymbol{\mu}). \tag{7}$$

Then, the transformer model [44], $\text{trans}(\cdot)$ predicts the subsequent latent vectors in an autoregressive manner.

$$\tilde{\boldsymbol{z}}_1 = \text{trans}(\boldsymbol{z}_{-1}, \boldsymbol{z}_0), \quad \tilde{\boldsymbol{z}}_t = \text{trans}(\boldsymbol{z}_{-1}, \boldsymbol{z}_0, \tilde{\boldsymbol{z}}_1, \ldots, \tilde{\boldsymbol{z}}_{t-1}) \tag{8}$$

Specifically, we use a single layer of multi-head attention in our transformer model, which we found sufficiently powerful for the environments we studied. We first describe the computation of a single attention head: At step $t$, the prediction $\tilde{\boldsymbol{z}}_{t-1}$ from the previous step issues a query to compute attention $\boldsymbol{a}_{(t-1)}$ over latents $(\boldsymbol{z}_{-1}, \boldsymbol{z}_0, \tilde{\boldsymbol{z}}_1, \ldots, \tilde{\boldsymbol{z}}_{t-1})$.

$$\boldsymbol{a}_{(t-1)} = \text{softmax}\left(\frac{\tilde{\boldsymbol{z}}_{t-1}^{\top} \boldsymbol{W}_1^{\top} \boldsymbol{W}_2}{\sqrt{d'}} \cdot \left[\boldsymbol{z}_{-1}, \boldsymbol{z}_0, \tilde{\boldsymbol{z}}_1, \ldots, \tilde{\boldsymbol{z}}_{t-1},\right]\right) \tag{9}$$

Here $\boldsymbol{W}_1$ and $\boldsymbol{W}_2$ are learnable parameters, and $d'$ denotes the length of the vector $\boldsymbol{z}_{t-1}$. This equation corresponds the popular inner-product attention model [44].

Next, the attention head computes the prediction vector

$$\boldsymbol{g}_t = \boldsymbol{W}_3\left[\boldsymbol{z}_{-1}, \boldsymbol{z}_0, \tilde{\boldsymbol{z}}_1, \ldots, \tilde{\boldsymbol{z}}_{(t-1)}\right]\boldsymbol{a}_{t-1}, \tag{10}$$

with the learned parameter $\boldsymbol{W}_3$.

Our multi-head attention model uses $K$ parallel attention heads with separate parameters to compute $K$ vectors $(\boldsymbol{g}_t^1, \ldots, \boldsymbol{g}_t^K)$ as described above. These vectors are concatenated and fed into an MLP to predict the residual of $\tilde{\boldsymbol{z}}_t$ over $\tilde{\boldsymbol{z}}_{t-1}$.

$$\tilde{\boldsymbol{z}}_t = \tilde{\boldsymbol{z}}_{t-1} + \text{mlp}_m\left(\text{concat}(\boldsymbol{g}_t^1, \ldots, \boldsymbol{g}_t^K)\right) \tag{11}$$

We train the autoregressive model on entire sequences by minimizing the prediction loss

$$\mathcal{L}_{att} = \sum_{t=1}^{T} \|\boldsymbol{z}_t - \tilde{\boldsymbol{z}}_t\|_2^2. \tag{12}$$

Compared to next-step models such as MeshGraphNet [35], this model can propagate gradients through the entire sequence, and can use information from all previous steps to produce stable predictions with minimal drift and error accumulation.

The proposed model also contains next-step models as special cases: if we select all nodes as pivotal nodes and set the temporal attention model such that it predicts the next step with only physical parameters and the current step, then the proposed model becomes a next-step model. The proposed model shows advantage when it reduces the dimension of representation by using pivotal nodes and expands the input range when predicting the next step. A.2 shows how this method reduces the amount of computation and memory usage.

### 3.4 MODEL TRAINING AND TESTING

We train the mesh reduction and recovery modules GMR, GMUS by minimizing equation 6 and the temporal attention model by minimizing equation 12. We obtain best results by training those components separately, as this provides stable targets for the autoregressive predictive module. A second reason is memory and computation efficiency; we can train the temporal attention method on full sequences in reduced space, without needing to jointly train the more memory-intensive GMR/GMUS modules.

The transformer model is trained by incrementally including loss terms computed from different time steps. For every time step $t$, we train the transformer by minimizing the objective $\sum_{t'=1}^{t} \| z_{t'} - \tilde{z}_{t'} \|_2^2$. Until the objective reaches a threshold, we add the next loss term $\| z_{t+1} - \tilde{z}_{t+1} \|$ to the objective and continue to train the model We found that this incremental approach leads to much more stable training than directly minimizing the overall objective.

As the transformer model can directly attend to all previous steps via equation 10, we don't suffer as much from vanishing gradients, compared to e.g. LSTM sequence models. By predicting in mesh-reduced space, the memory footprint is limited, and the model can be trained without approximate gradient calculations such as teacher forcing or limiting the attention history. Details for training procedure can be found in section A.3.

Once we have trained our models, we can make predictions for a new problem. We first represent system parameters as a token $z_{-1}$ by equation 7 and encode the initial state into a token $z_0 = \text{GMR}(\mathcal{G}, Y_0)$, then we predict the sequence $\tilde{z}_1, \ldots, \tilde{z}_T$ by equation 8, and finally we decode the system state $\tilde{Y}$ by

$$\tilde{Y}_t = \text{GMUS}(\tilde{z}_t), \quad t = 1, \ldots, T. \tag{13}$$

## 4 RESULTS

**Datasets.** We tested the proposed method to three datasets of fluid dynamic systems: (1) flow over a cylinder with varying Reynolds number; (2) high-speed flow over a moving wedge with different initial temperatures; (3) flow in different vascular geometries. These datasets correspond to applications with fixed, moving, and varying unstructured mesh, respectively. A detailed description of the datasets can be found in section A.4.

**Methods.** The proposed methods is compared to the state-of-the-art MeshGraphNet method [35], which is a next-step model and has been shown to outperform a list of previous methods. Two versions of MeshGraphNet are considered here, with and without Noise Injection (NI). We also study variants of our model with LSTM and GRU (GMR-LSTM and GMR-GRU) instead of temporal attention. These variants also operate in mesh-reduced space, and use the same encoder (GMR) and decoder (GMUS) as our main method. Model details can be found in section A.5. And section A.6 shows pivotal nodes for each dataset.

**Stable and visually accurate model predictions.** We tested the proposed method on three fluid systems. Figure 2 compares our model's predictions for velocity against the ground truth (CFD) on six test set trajectories. Our model shows stable predictions even without noise injection, and model rollouts are visually very closely to the CFD reference. On the other hand, without mitigation strategies, next-step models often struggle to predict such long sequences stably and accurately. For cylinder flow, our model can successfully learn the varying dynamics under different Reynolds numbers, and accurately predict their corresponding transition behaviors from laminar to vortex shedding. In supersonic flow over a moving wedge, our model can accurately capture the shock bouncing back and forth on a moving mesh. In particular, the small discontinuities at the leeward side of the edge are discernible in our model predictions. Lastly, in vascular flow with a variable-size circular thrombus, and thus different mesh sizes, the model is able to predict the flow accurately. In both cylinder flow and vascular flow, both frequency and phase of vortex shedding is accurately captured. The excellent agreement between model predictions and reference simulation demonstrates the capability of our model when predicting long sequences with varying-size meshes. Results for pressure can be found in section A.9.

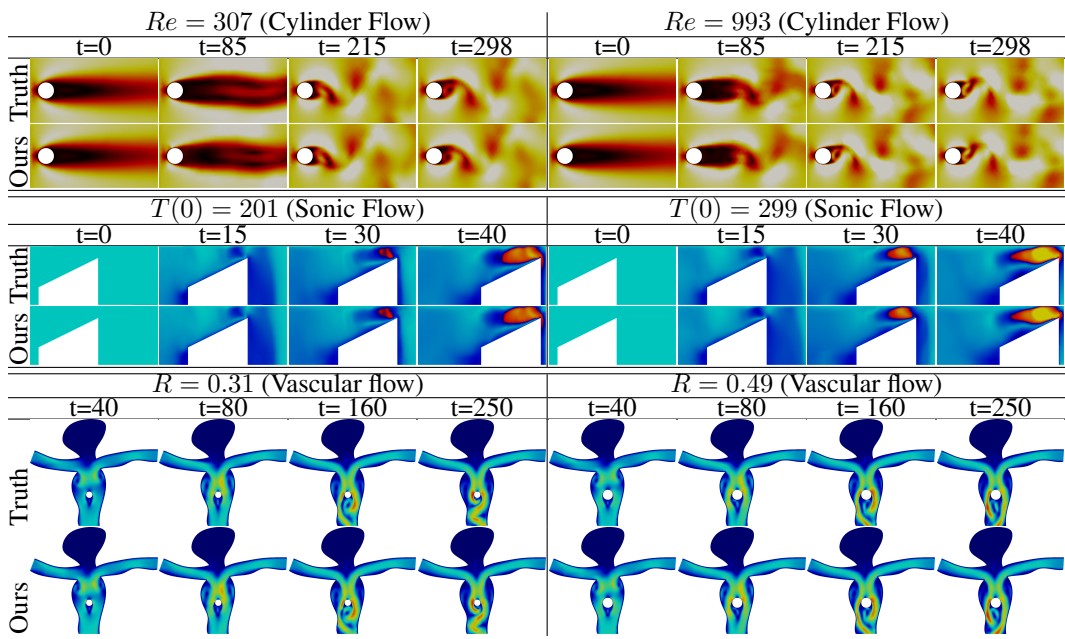

Figure 2: Contours of the velocity field, as predicted by our model versus the ground truth (CFD). Our model accurately predicts long rollout sequences under varying system parameters.

Table 1: The average relative rollout error of three systems, with unit of $\times 10^{-3}$. We compare MeshGraphNet with or without noise injection (NI), to three variants of the our model (LSTM, GRU, or Transformer), which all shared the same GMR/GMUS mesh reduction models. Our model significantly outperform MeshGraphNet on datasets with long rollouts.

| Dataset-rollout step | | Cylinder flow-400 | | | Sonic flow-40 | | | | Vascular flow-250 | | |
|---|---|---|---|---|---|---|---|---|---|---|---|
| Variable | | $u$ | $v$ | $p$ | $u$ | $v$ | $p$ | $T$ | $u$ | $v$ | $p$ |
| MeshGraphNet | NI | 25 | 778 | 136 | 1.71 | 3.67 | 0.4 | 0.027 | 57 | 133 | 55 |
| | without NI | 98 | 2036 | 673 | 4.12 | 6.13 | **0.24** | **0.020** | 3117 | 1771 | 601 |
| Ours | GRU | 114 | 1491 | 1340 | 1.34 | 4.59 | 0.59 | 0.37 | 8.2 | 11.2 | 23.6 |
| | LSTM | 124 | 1537 | 1574 | 1.57 | 5.8 | 0.69 | 0.45 | 8.4 | 11.1 | 23.3 |
| | Transformer | **4.9** | **89** | **38** | **0.95** | **2.8** | 0.43 | 0.39 | **7.3** | **10** | **22** |

**Error behavior under long rollouts** To quantitatively assess the performances of our models and baselines, we compute the relative mean square error (RMSE), defined as $\mathrm{RMSE}(\hat{\mathbf{u}}^{\mathrm{prediction}}, \hat{\mathbf{u}}^{\mathrm{truth}}) = \frac{\sum(\hat{u}_i^{\mathrm{prediction}} - \hat{u}_i^{\mathrm{truth}})^2}{\sum(\hat{u}_i^{\mathrm{prediction}})^2}$, over the full rollout trajectory. Table 1 compares average prediction errors on all three domains. Our attention-based model outperforms baselines and model variants in most scenarios. For sonic flow, which has 40 time steps, our model performs better than MeshGraphNet when predicting velocity values ($u$ and $v$). The quantities $p$ and $T$ are relatively simple to predict, and all models show very low relative error rates ($< 10^{-3}$), though our model has slightly worse performance. The examples with long rollout sequences (predicting cylinder flow and vascular flow) highlight the superior performance of our attention-based method compared with baseline next-step models, and the RMSE of our method is significantly lower than baselines.

Figure 3 shows how error accumulates in different models. Our model has higher error for the first few iterations, which can be explained by the information bottleneck of mesh reduction. However, the error only increases very slowly over time. MeshGraphNet on the other hand suffers from strong error accumulation, rendering it less accurate for long-span predictions. Noise injection partially addresses the issue, but error accumulation is still considerably higher compared to our model. We can attribute this better long-span error behavior of our sequence model, to being able to pass gradients to all previous steps, which is not possible in next-step models. We also note that the

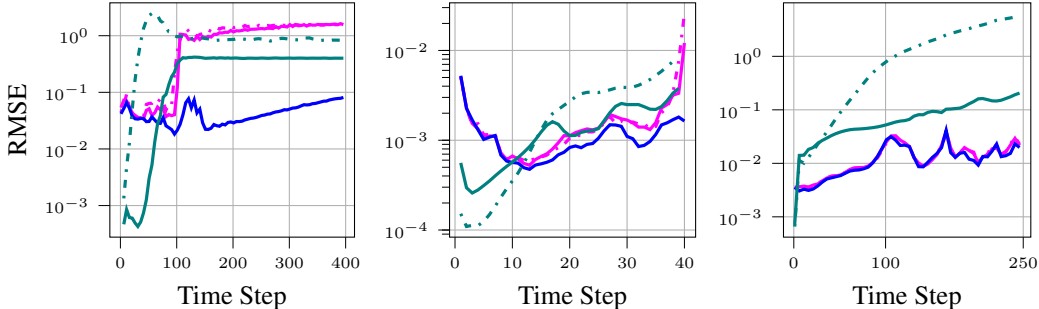

Figure 3: Averaged error over all state variables on cylinder flow (left), sonic flow (middle) and vascular flow (right), for the models MeshGraphNets(MGN)(· - · -), MGN-NI(———), Ours-GRU (———), Ours-LSTM (· - · -), Ours-Transformer (———). Our model, particularly the transformer, show much less error accumulation compared to the next-step model.

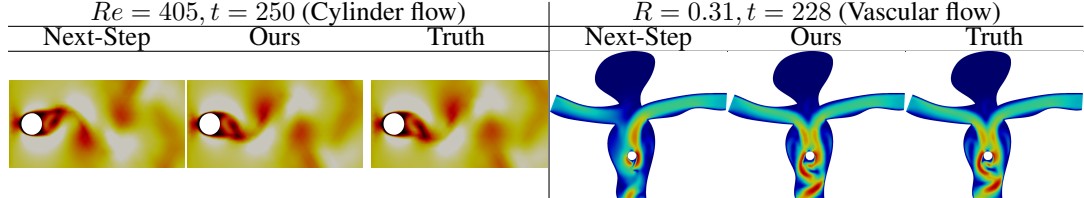

Figure 4: Predictions of the next-step MeshGraphNet model and our model, compared to ground truth. The next-step model fails to keep the shedding frequency and show drifts on cylinder flow. On vascular flow, we notice the left inflow diminishing over the time, while our model remains close to the reference simulation.

transformer model performs much better compared to the GRU and LSTM model variants. We believe its ability to directly attend to long-term history helps to stabilize gradients in training, and also to identify recurrent features of the flow, as discussed later in this section.

In Figure 4 we illustrate the difference in the models' abilities to predict long rollouts. For cylinder flow, our model is able to retain phase information. MeshGraphNet on the other hand drifts to an opposite vortex shedding phase at $t = 250$, though its prediction still looks plausible. For vascular flow, our model gives an accurate prediction at step $t = 228$, while the prediction of MeshGraphNet is significantly different from the ground-truth at this step: the left inflow diminishes, making the prediction less physically plausible.

**Diagnosis of state latents.** The low-dimensional latent vectors, computed by the encoder-decoder (GMR and GMUS), can be viewed as lossy compression of high-dimensional physical data on irregular unstructured meshes. In our experiment, we observe the compression ratio is $\leq 4\%$ while the loss of the information measured by RMSE is below $1.5\%$. More details are in section A.7.

We also visualize state latents **z**-s from the cylinder flow dataset and show that the model can distinguish rollouts from different Reynolds numbers. We first use Principal Component Analysis (PCA) to reduce the dimension of **z**-s to two and plot them in Figure 5 (left). Rollouts with two different Reynolds numbers have two clearly distinctive trajectories. They eventually enters their respective periodic phase. We also apply PCA directly to the system states (**Y**) and get 2-d latent vectors directly from the original data. We plot these vectors in Figure 5 (right). Latent vectors from PCA start in the center and form a circle. The two trajectories from two different Reynolds numbers are hard to distinguish, which makes it challenging for the learned model to capture parameter variations. Although CNN-based embedding methods [15, 51, 30] are also nonlinear, they cannot handle irregular geometries with unstructured meshes due to the limitation of classic convolution operations. Using pivotal nodes combined with GNN learning, the proposed model is both flexible in dealing data with irregular meshes and effective in capturing state transitions in the system.

**Diagnosis of attention weights.** The attention vector $\boldsymbol{a}_t$ in equation 9 plays an important role in predicting the state of the dynamic system. For example, if the system enters a perfect oscillating

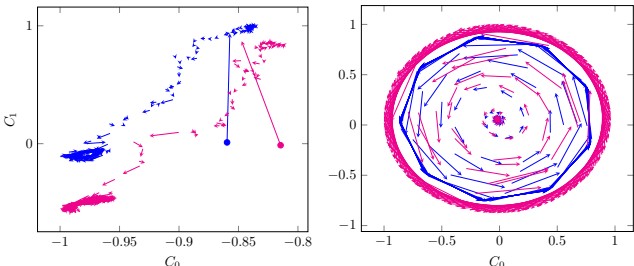 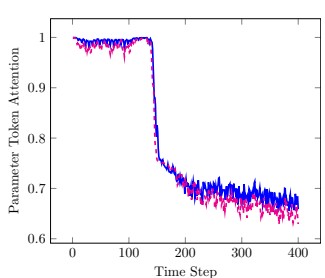

Figure 5: 2-D principle subspace of the latent vectors from GMR (left) and PCA (middle) for flow past cylinder system: $Re = 307$ (——), $Re = 993$ (——) start from ——• and ——• respectively.

Figure 6: Attention values of the parameter tokens versus time, $Re = 307$ (——), and $Re = 993$ (- - -).

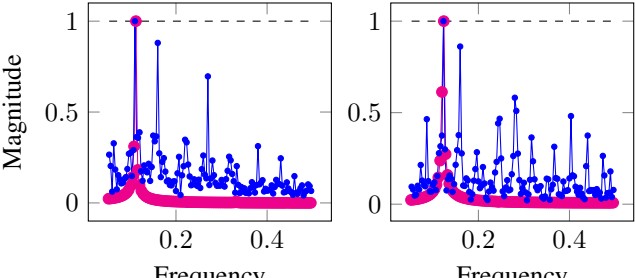 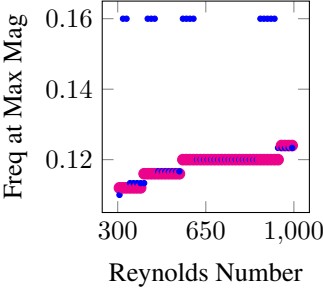

Figure 7: Attention (——•—), CFD data (——•—). (left) $Re$ versus frequency for attention and CFD data. Fourier transform for both attention and CFD data at (middle) $Re = 307$ and (right) $Re = 993$.

stage, the model could directly look up and re-use the encoding from the last cycle as the prediction, to make predictions with little deterioration or drift.

We can observe the way our model makes predictions via the attention magnitude over time. Figure 6 shows attention weights for the initial parameter token $z_{-1}$, which encodes the system parameters $\mu$, over one cylinder flow trajectory. In the initial transition phase, we see high attention to the parameter token, as the initial velocity fields of different Reynold numbers are hard to distinguish. As soon as vortex shedding begins around $t = 120$, and the flow enters an oscillatory phase, and attention shift away from the parameter token, and locks onto these cycles. This can be seen in Figure 7 (left): we perform a Fourier analysis of the attention weights, and notice that the peak of the attention frequency (blue) coincides with the vortex shedding frequency in the data (purple). That is, our dynamics model learned to attend to previous cycles in vortex shedding, and can use this information to make accurate predictions that keep both frequency and phase stable. Figure 7 (right) shows that this observation holds for the whole range of Reynolds numbers, and their corresponding vortex shedding frequencies.

For sonic flow, the system state during the first few steps is much easier to discern even without knowing the system parameters $\mu$, and hence, the attention to $z_{-1}$ quickly decays. This shows that the transformer can adjust the attention distribution to the most physically useful signals, whether that is parameters or system state. Analysis on this, as well as the full attention weights maps can be found in section A.10.

## 5 CONCLUSION

In this paper we introduced a graph-based mesh reduction method, together with a temporal attention module, to efficiently perform autoregressive prediction of complex physical dynamics. By attending to whole simulation trajectories, our method can more easily identify conserved properties or fundamental frequencies of the system, allowing prediction with higher accuracy and less drift compared to next-step methods.

## ACKNOWLEDGEMENT

Han Gao and Jianxun Wang are supported by the National Science Foundation under award numbers CMMI-1934300 and OAC-2047127. Li-Ping Liu are supported by NSF 1908617. Xu Han was also supported by NSF 1934553.

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

# A APPENDIX

## A.1 INFORMATION FLOW IN GMR AND GMUS

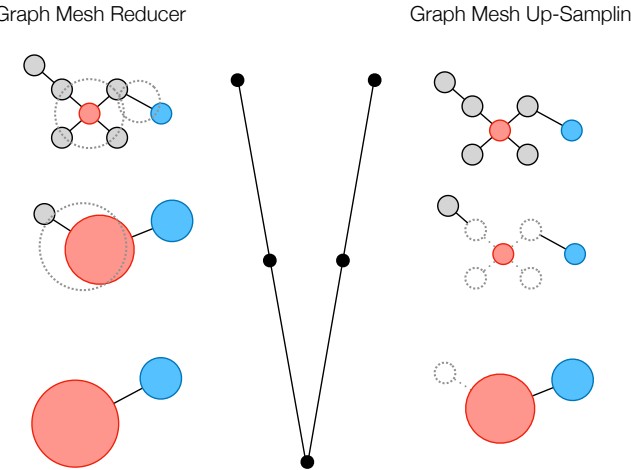

Figure 8: Schematic of information flow in the Graph Mesh Reducer (GMR) and Graph Mesh Up-Sampling (GMUS). Red and blue nodes are pivotal nodes. The node size increase represents information aggregation from surroundings.

Figure 8 shows how the information flows in Graph Mesh Reducer (GMR) and Graph Mesh Up-Sampling (GMUS). This is analogous to a **V-cycle** in multigrid methods: Consider a graph with seven nodes ($|\mathcal{V}| = 7$) to be reduced and recovered by GMR and GMSU with two message-passing layers. The increased size of pivotal nodes after a message-passing layer represents the information aggregation from surrounding nodes.

## A.2 COMPUTATIONAL COMPLEXITY

The computation of the proposed model is from the three components: GMR, GMUS, and the transformer. The graph $\mathcal{G}$ contains $N$ nodes and also has $O(N)$ edges because of its special structure. Then GMR and GMUS as $L$-layer graph neural networks have computation time $O(NLd^*)$, with $d^*$ being the maximum number of hidden units.

The transformer takes time $O(T \cdot (d')^2)$ to make a prediction: a prediction needs to consider $O(T)$ previous steps, and the computation over a time step is $O((d')^2)$. Note that $d'$ is the length of the latent vector at each step.

In our actual calculation, the transformer takes much less time than GMR or GMUS since the latent dimension $d'$ is small.

However, it would be very expensive if the model did not use GMR and directly applied a transformer to the original system states. In this case, $d'$ would be $Nd$, and the total complexity would be $O(TN^2d^2)$, which would not be feasible for typical computation servers.

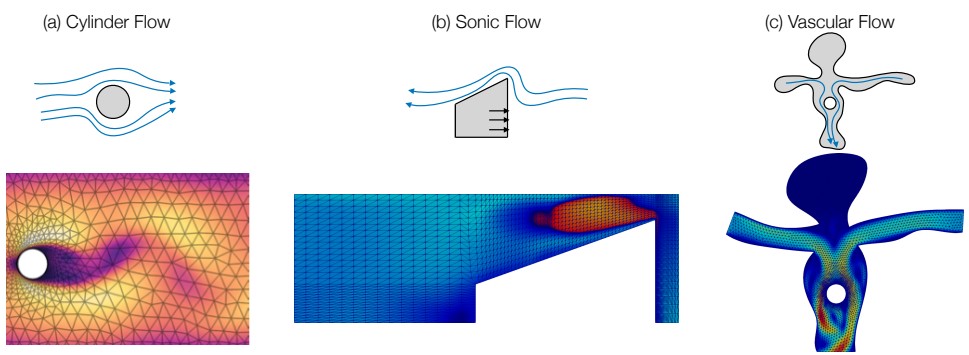

Figure 9: Overview of datasets of (a) cylinder flow, (b) sonic flow, and (c) vascular flow.

## A.3 MODEL TRAINING

The GMR/GMUS and temporal attention model are trained separately. We first train GMR/GMUS by minimizing the reconstruction loss $\mathcal{L}_{graph}$ defined by Eqn. 6. Then, the parameters of the trained encoder-decoder are frozen, and only the parameters of the temporal attention model will be trained by minimizing the prediction loss $\mathcal{L}_{att}$ (Eqn. 12) with early stopping. The parameter token encoder $\text{mlp}_p$ is trained jointly with temporal attention. The training algorithm is detailed in Algorithm 1.

---

**Algorithm 1** Training Process

**Input:** Domain graph $\mathcal{G}$, Node features over time $[\boldsymbol{Y}_0, \ldots, \boldsymbol{Y_T}]$, $\text{GMR}_\theta$, $\text{GMUS}_\phi$, $\text{Attention}_\omega$, RMSE threshold $R'$, $\text{MLP}_\psi$, condition parameter $\mu$
**Output:** Learned parameters $\theta$, $\phi$, $\omega$ and $\psi$
**repeat**
    **for** $\boldsymbol{Y}_t \in [\boldsymbol{Y}_0, \ldots, \boldsymbol{Y}_T]$ **do**
        $\hat{\boldsymbol{Y}}_t = \text{GMUS}_\phi(\text{GMR}_\theta(\boldsymbol{Y}_t, \mathcal{G}))$
    **end for**
    Compute $\nabla_{\theta,\phi} \leftarrow \nabla_{\theta,\phi} \mathcal{L}_{graph}(\theta, \phi, [\boldsymbol{Y}_0, \ldots, \boldsymbol{Y}_T], [\hat{\boldsymbol{Y}}_0, \ldots, \hat{\boldsymbol{Y}}_T])$
    Update $\phi$, $\theta$ using the gradients $\nabla_\phi$, $\nabla_\theta$
**until** convergence of the parameters $(\theta, \phi)$
$[z_0, z_1, \ldots, z_T] = \text{GMR}_\theta([\boldsymbol{Y}_0, \ldots, \boldsymbol{Y}_T])$, $z_{-1} = \text{MLP}_\psi(\mu)$
**for** $t \in [1, 2, \ldots, T]$ **do**
    **repeat**
        $[\hat{z}_1, \ldots, \hat{z}_t] = \text{Attention}_\omega([z_{-1}, z_0])$
        Compute $\nabla_{\omega,\psi} \leftarrow \nabla_{\omega,\psi} \mathcal{L}_{att}(\omega, \psi, [\boldsymbol{z}_1, \ldots, \boldsymbol{z}_t], [\hat{\boldsymbol{z}}_1, \ldots, \hat{\boldsymbol{z}}_t])$
        Update $\omega$, $\psi$ using the gradients $\nabla_\omega$, $\nabla_\psi$
    **until** $\text{RMSE}([\boldsymbol{z}_1, \ldots, \boldsymbol{z}_t], [\hat{\boldsymbol{z}}_1, \ldots, \hat{\boldsymbol{z}}_t]) < R'$
**end for**

---

## A.4 DATASET DETAILS

Three flow datasets, cylinder flow, sonic flow and vascular flow, are used in our numerical experiments, and the schematics of flow configurations are shown in Figure 9. Cylinder and vascular flow are governed by incompressible Navier-Stokes (NS) equations, whereas compressible NS equations govern sonic flow. The ground truth datasets are generated by solving the incompressible/compressible NS equations based on the finite volume method (FVM). We use the open-source FVM library OpenFOAM [20] to conduct all CFD simulations. All flow domains are discretized by triangle (tri) mesh cells and the mesh resolutions are adaptive to the richness of flow features. The case setup details can be found in Table 2. Both cylinder flow and sonic flow datasets consist of 51 training and 50 test trajectories, while vascular flow consists of 11 training and 10 test trajectories.

Table 2: Simulation details of datasets

| Dataset | PDEs | Cell type | Meshing | # nodes | # steps | $\delta t$ (sec) |
|---|---|---|---|---|---|---|
| Cylinder flow | Incompr. NS | tri | Fixed | 1699 | 400 | 0.5 |
| Sonic flow | Compr. NS | tri | Moving | 1900 | 40 | $5 \times 10^{-7}$ |
| Vascular flow | Incompr. NS | tri | Small-varying | 7561 (avg.) | 250 | 0.5 |

Table 3: Input-output information of our networks: $u, v, p, T, \rho, m$ denote X-axis velocity, Y-axis velocity, pressure, temperature, density and cell volume, respectively. $\mathbf{x}$ denote the cell coordinates. $T_{,0}, Re, r$ denote the initial temperature (sonic flow), Reynolds number (cylinder flow) and radius of the thrombus (vascular flow).

| Dataset | GMR node input | GMR edge input | GMUS node output | Parameter token model input | Nodal embed dim | # selected node |
|---|---|---|---|---|---|---|
| Cylinder flow | $u_i, v_i, p_i, m_i, Re$ | $\boldsymbol{x}_i - \boldsymbol{x}_j,$ $\|\boldsymbol{x}_i - \boldsymbol{x}_j\|$ | $u_i, v_i, p_i$ | $Re$ | 4 | 256 |
| Sonic flow | $u_i, v_i, p_i, T_i, \rho_i$ $m_i, T_{,0}$ | $\boldsymbol{x}_i - \boldsymbol{x}_j,$ $\|\boldsymbol{x}_i - \boldsymbol{x}_j\|$ | $u_i, v_i, p_i$ $T_i, \rho_i$ | $T_{,0}$ | 4 | 256 |
| Vascular flow | $u_i, v_i, p_i, m_i, r$ | $\boldsymbol{x}_i - \boldsymbol{x}_j,$ $\|\boldsymbol{x}_i - \boldsymbol{x}_j\|$ | $u_i, v_i, p_i$ | $r$ | 2 | 400 |

Here we summarize all input and outputs of the GMR/GMUS models, as well as the attention simulator for each dataset. GMR encodes the high-dimensional system state onto the node presentations on pivotal nodes, while GMR decodes latents back to the high-dimensional physical state (Figure 8). The parameter token encoder takes the physical parameter $\mu$ as input, and outputs a parameter token. The attention model takes the parameter token, together with the latent vectors of states at all previous time steps $\mathbf{z}$, to predict the latents of the next time step. All input and output variables for each dataset are detailed in Table 3.

The governing equations for all fluid dynamic cases can be summarized as follows,

$$\frac{\partial \rho}{\partial t} + \nabla \cdot (\rho \boldsymbol{v}) = 0,$$

$$\frac{\partial (\rho \boldsymbol{v})}{\partial t} + \nabla \cdot (\rho \boldsymbol{v} \boldsymbol{v}) = \nabla \cdot (\mu \nabla \boldsymbol{v}) - \nabla p + \nabla \cdot (\mu (\nabla \boldsymbol{v})^T) - \frac{2}{3} \nabla (\mu \nabla \cdot \boldsymbol{v}), \quad (14)$$

$$\frac{\partial}{\partial t} (\rho c_p T) + \nabla \cdot (\rho c_p \boldsymbol{v} T) = \nabla \cdot (k \nabla T) + \rho T \frac{D c_p}{D t} + \frac{D p}{D t} - \frac{2}{3} \mu \Psi + \mu \Phi,$$

where $\boldsymbol{v}$ is the velocity vector, $\mu$ is the viscosity, $k$ is the thermal conductivity, $c_p$ is the specific heat of the fluid and the definitions of the stream scalar $\Psi$, the potential scalar $\Phi$ are referred to the chapter 3 in [29]. Only the sonic flow (compressible flow) involves solving the third equation.

For cylinder flow, the simulation mesh topology remains the same for all trajectories, and the Reynolds number varies. In sonic flow, topology also remains fixed; however, the mesh node coordinates change over the course of the simulation to simulate the moving ramp. Here, we vary the initial temperature. Finally, for vascular flow, we vary the radius of the thrombus, which means that each trajectory will have a different mesh topology, and the simulation model should be able to work with variable-sized inputs in this case.

## A.5 MODEL DETAILS

Node and edge representations in our GraphNets are vectors of width 128. The node and edge functions ($\text{mlp}_v$, $\text{mlp}_e$, $\text{mlp}_r$) are MLPs with two hidden layers of size 128, and ReLU activation. The input dimension of $\text{mlp}_v$ is based on the number of input features in each node (see details in table 3), and the input dimension of $\text{mlp}_e$ is three due to the one-layer neighboring edge. We use $L = 3$ GraphNet blocks for both GMR and GMUS. The node and edge functions $\text{mlp}_\ell^e$ and $\text{mlp}_\ell^v$ used in the GraphNet blocks are 3-Layer, ReLU-activated MLPs with hidden size of 128. The output

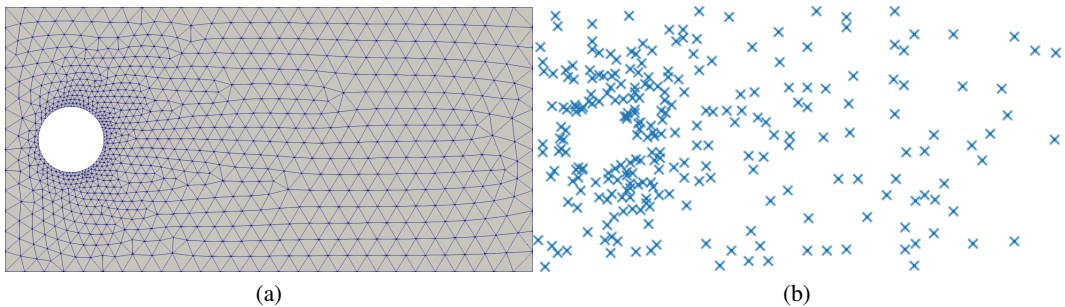

Figure 10: Flow past cylinder flow system: (a) all cells in the FV mesh; (b) pivotal nodes

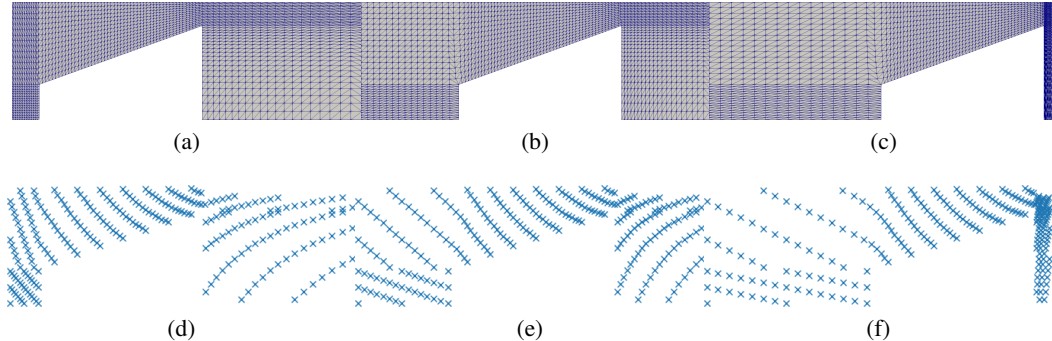

Figure 11: Sonic flow system: (a-c) all cells in the FV mesh for 3 different time steps; (d-f) pivotal nodes for 3 different time steps.

size is 128 for the all MLPs, except for the final layer, which is of size 4 for Cylinder, Sonic and 2 for Vascular.

We use a single layer and four attention heads in our transformer model. The embedding sizes of $z$ for each dataset (cylinder flow, sonic flow, and vascular flow) are 1024, 1024, and 800, respectively. This is calculated by #pivotal nodes $\times n_{out}$, in which $n_{out}$ is the output size of the last GraphNet block. The $\text{mlp}_p$ used to encode the system parameter $\mu$ is a MLP with 2 hidden layers of size 100, and with output dimensions that match $z$, i.e. 1024, 1024, and 800 for Cylinder Flow, Sonic Flow, and Vascular Flow, respectively.

Table 4: Number of parameters for each model

| Dataset | MeshGraphNet | GRU | LSTM | Transformer | GMR-GMU |
|---|---|---|---|---|---|
| Cylinder flow | 2.2M | 9.4M | 11.5M | 14.1M | 1.2M |
| Sonic flow | 2.2M | 9.4M | 11.5M | 14.1M | 1.2M |
| Vascular flow | 2.2M | 5.8M | 7.0M | 8.5M | 1.2M |

### A.6 PIVOTAL POINTS SELECTION

In general, pivotal nodes are selected by uniformly sampling from the entire mesh, as it preserves the mesh density distribution, which is designed based on the flow physics observed in training data. For our datasets, we select 256 pivotal nodes out of 1699 cells for cylinder flow, 256 pivotal nodes out of 1900 cells for sonic flow, and 400 pivotal nodes out of 7561 cells for vascular flow. The pivotal nodes in vascular flow are manually reduced in the aneurysm region, where flow features are not rich. The pivotal nodes distributions for the three flow systems are shown in Figures 10, 11, and 12, respectively.

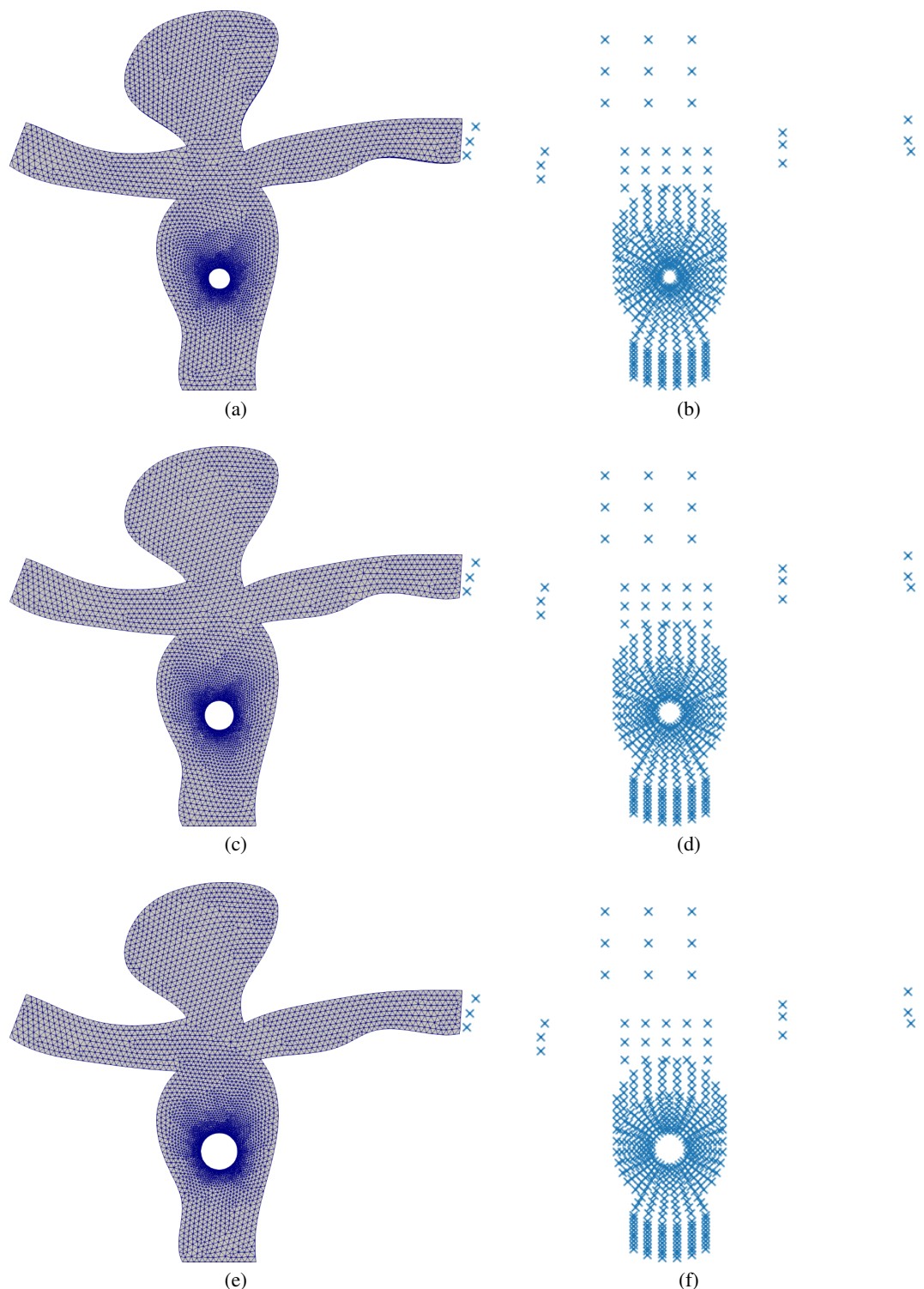

Figure 12: Vascular flow system: all cells in the FV mesh (left), pivotal nodes (right), $r = 0.3$ (a-b), $r = 0.4$ (c-d), $r = 0.5$ (e-f).

Table 5: Report of data compressing: QoI data represent original data of the variables of interest; Embedding represents the saved embedding vectors from the GMR (encoder); Decoder is the saved GMUS model which is used to decode the embeddings; Compression Ratio is the ratio of QoI data size to the sum of the embedding and decoder sizes. The RMSE is the relative mean square error of the reconstructed data from embedding.

| Dataset | QoI Data | Embedding | Decoder | Compression Ratio | RMSE ($\times 10^{-3}$) |
|---|---|---|---|---|---|
| Cylinder flow | 3570 MB | 82.1MB | 12.8MB | **37** | 14.3 |
| Sonic flow | 651MB | 8.5MB | 18 MB | **25** | 1.11 |
| Vascular flow | 1951MB | 8.5MB | 49.3 MB | **33** | 10 |

## A.7    DATA COMPRESSION

The Graph Mesh Reducer (GMR) and Graph Mesh Up-Sampling (GMUS) can be treated as an encoder and decoder for compressing high-dimensional physical data on irregular unstructured meshes. For any given mesh data $(\mathcal{G}, \boldsymbol{Y})$, we can encode it into a latent vector $\boldsymbol{z} = \text{GMR}(\mathcal{G}, \boldsymbol{Y})$ through GMR. Therefore, a high-dimensional spatiotemporal trajectory $(\mathcal{G}, \boldsymbol{Y}_0, \ldots, \boldsymbol{Y}_T)$ can be compressed into a low-dimensional representation $(\mathcal{G}, \boldsymbol{z}_0, \ldots, \boldsymbol{z}_T)$, which significantly reduces memory for storage. Similar to other compressed sensing techniques, the original data can be restored by GMUS from latent vectors. Table 5 shows that the decoder can significantly reduce the size of the original data to about 3% with a slight loss of accuracy after being restored by the decoder. Note that only the GMR/GMUS are included in the data compressing process, the transformer dynamics model is not involved.

## A.8    INFERENCE TIME

Our ground truth solver uses the finite volume method to solve the governing PDEs. Specifically, the "pressure implicit with splitting of operators" (PISO) algorithm and "semi-implicit method for pressure linked equations" (SIMPLE) [29] are used to simulate the unsteady cylinder and vascular flows, and a PISO-based compressible solver [26] is used to simulate the high-speed sonic flow. All these traditional numerical methods involve a large number of iterations and time integration steps. In contrast, the trained neural network model is able to make predictions without the need of sub-timestepping. Hence, these predictions can be are significantly faster than running the ground-truth simulator. The online evaluation of our learned model at inference time only involves two steps,

$$\text{Step 1:} \quad \hat{\boldsymbol{z}}_1 = \text{trans}(\boldsymbol{z}_{-1}, \boldsymbol{z}_0), \quad \hat{\boldsymbol{z}}_t = \text{trans}(\boldsymbol{z}_{-1}, \boldsymbol{z}_0, \hat{\boldsymbol{z}}_1, \ldots, \hat{\boldsymbol{z}}_{t-1})$$
$$\text{Step 2:} \quad \tilde{\boldsymbol{Y}}_t = \text{GMUS}(\tilde{\boldsymbol{z}}_t), \quad t = 1, \ldots, T$$

(15)

which are completely decoupled. We compare the evaluation cost of the learned model with the FV-based numerical models in Table 6, and observe significant speedups for all three datasets.

Table 6: Wall-clock time of finite volume (FV) model and attention-GMUS to simulate a trajectory for three flow cases.

| Dataset | GMUS | Transformer | GT Model | Speed-up |
|---|---|---|---|---|
| Cylinder flow (for 400 time steps) | 2sec | 0.4758sec | 1688.59sec | **682** |
| Sonic flow (for 40 time steps) | 0.2sec | 0.047sec | 25.026sec | **100** |
| Vascular flow (for 250 time steps) | 2.63sec | 0.31sec | 2451sec | **800** |

## A.9    OTHER VARIABLE CONTOUR

Figure 13 shows the pressure contours.

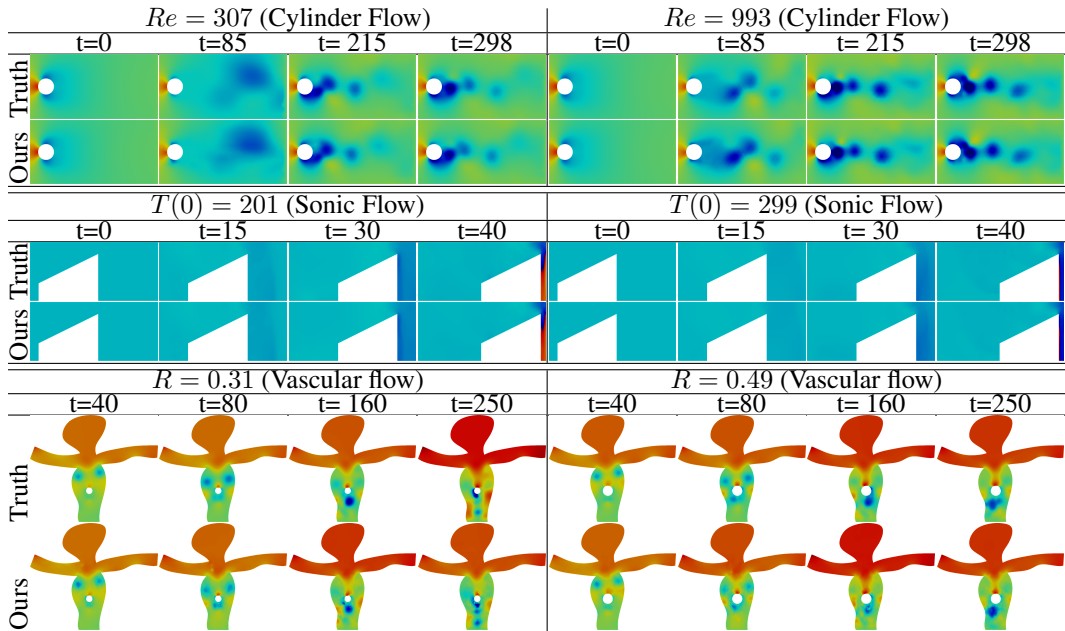

Figure 13: The pressure contours of rollouts predicted by our model versus the ground truth (CFD). With different condition settings, the model accurately predict long sequences of rollouts.

## A.10 FURTHER ANALYSIS OF LATENT ENCODINGS AND ATTENTION

Figure 14 analyzes the latent encoding and attention for the sonic flows in a similar vein to section 4. Different initial temperatures (as the system parameter) lead to fast diverging of state dynamics, and thus the parameter tokens overlap for different parameters. However, the attentions to $\mathbf{z}_{-1}$ also rapidly decay, demonstrating that the transformer can adjust the attention distribution to capture most physically useful signals.

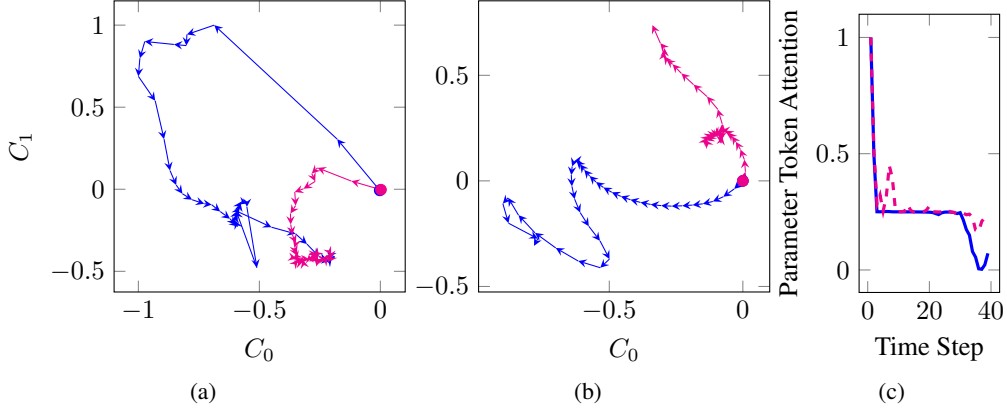

Figure 14: 2-D principle subspace of the embedded vectors from GMR (a) and PCA (b) for the high-speed sonic flow system: $T(0) = 201$ trajectory ($\longrightarrow$), $T(0) = 299$ trajectory ($\longrightarrow$) start from their parameter tokens $\bullet$ and $\bullet$, respectively. (c) Attention values of the parameter tokens versus time, $T(0) = 201$ ($\longrightarrow$), and $T(0) = 299$ ($\text{- - -}$).

Figure 15 shows the overview of attention value distribution in a time-time diagram for all three fluid systems. The attentions of the state to that at different time step is plotted as a log-scale contour. Note that the sum of attention values in the same row equals to 1 due to the softmax layer (Eqn. 9).

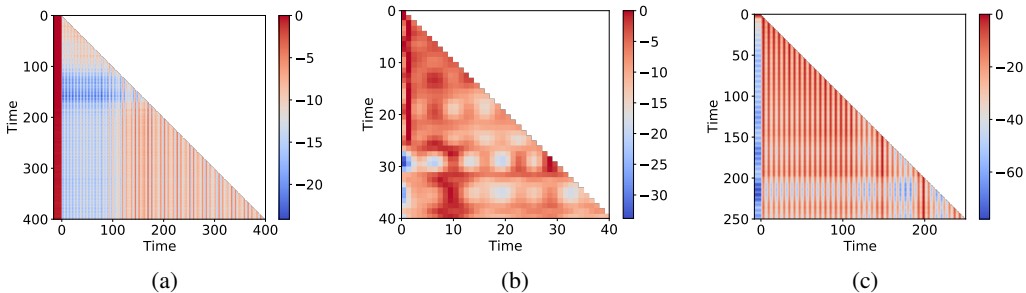

|       (a)       |       (b)       |       (c)       |

Figure 15: Examples of attention value (log value contour) for (a) flow past cylinder, (b) sonic flow and (c) vascular flow.

Table 7: The average relative rollout error for the cylinder flow, with unit of $\times 10^{-3}$. We compare MeshGraphNet with noise injection (NI) to our model with transformer. We train the model on the training trajectories with 400 steps, and test it on the unseen trajectories with 800 steps.

| rollout step | 800 | | |
|---|---|---|---|
| Variable | $u$ | $v$ | $p$ |
| MeshGraphNet-NI | 43 | 1274 | 258 |
| Ours-Transformer | **6** | **158** | **48** |

## A.11 LONGER ROLLOUTS

We perform an additional experiment to test how the model performs when rolling out of the training range. We train a model for the cylinder flow setup ($n$=400 steps), and evaluate it for lengths of $2n$.

To limit the memory footprint for very long rollouts, we use a sliding window of $n$ for the attention, i.e., we always attend to the last $n$ time steps. This way allow us to retain the same complexity for the training. Fig. 16 illustrates this procedure: we always use the parameter embedding token $\tilde{z}_{-1}$ as the first entry of the attention sequence, followed by a moving window of the $n$ last timesteps.

Table 7 compares the performance of the proposed model with MeshGraphNet on a longer rollout length ($2n = 2 \times 400$ steps). We find that the proposed model still outperforms the baseline for all $u, v$ and $p$ trajectories.

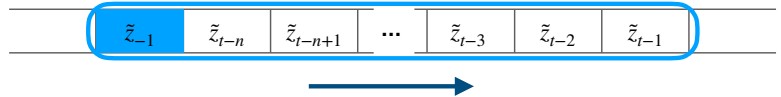

Figure 16: Sliding window for attention mechanism on latent representations. The parameter embedding token $\tilde{z}_{-1}$ is always included in the first entry.

## A.12 PREDICTION ON CONVECTION-DIFFUSION FLOW

Many of the systems we study in the main paper have oscillatory behaviors, which are hard to capture by next-step models stably. To further demonstrate the capability of the model on predicting non-oscillatory dynamics, we add an additional example: a convection-diffusion flow, governed by

$$\frac{\partial c}{\partial t} + \nabla \cdot (\boldsymbol{v}c) = \nabla^2 (\frac{|\boldsymbol{v}|}{Pe}c), \tag{16}$$

where $c$ is the concentration of the transport scalar, $\boldsymbol{v} = [1, 1]$ is the convection velocity, and $Pe$ is the Peclet number, which controls the ratio of the advective transport rate to its diffusive transport rate.

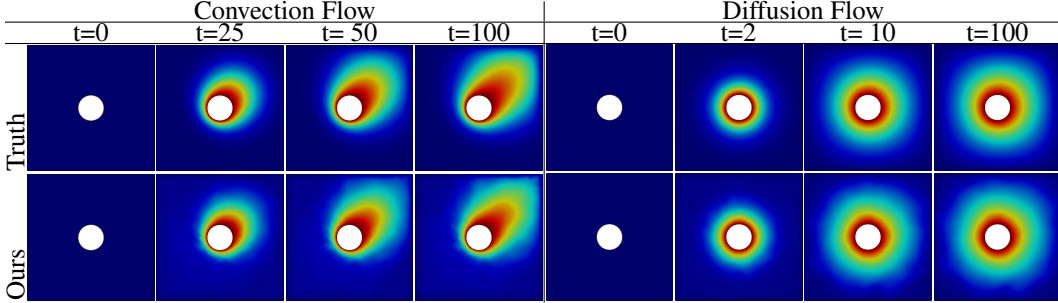

Figure 17: The concentration contours of rollouts predicted by our model versus the ground truth (CFD).

As shown in Figure 17, the model can accurately predict the non-oscillatory sequences of rollouts Quantitatively, our model (RMSE $= 1.03 \times 10^{-3}$) still outperforms MeshGraphNet (RMSE $= 2.79 \times 10^{-3}$).

### A.13 CARDIAC FLOW PREDICTION

We test the proposed model on cardiac flows (pulsatile flows) with varying viscosity in multiple cardiac cycles, which is more realistic for cardiovascular systems. An idealized spatial-temporal inflow condition is defined as

$$v(x,t) = \frac{0.6(x - x_{\min})(x - x_{\max})}{(x_{\max} - x_{\min})^2}(\sin(\frac{2\pi t}{1.8}))^2 + 0.15 \tag{17}$$

to simulate the pulsating inflow from the heart. The model can accurately predict 10 cardiac cycles (Figure 18). The comparison with MeshGraphNet is listed in Table 8, showing significantly better performance for our model.

Table 8: The average relative rollout error for cardiac flow, with unit of $\times 10^{-3}$.

| rollout step | 300 | | |
|---|---|---|---|
| Variable | $u$ | $v$ | $p$ |
| MeshGraphNet-NI | 194 | 71 | 944 |
| Ours-Transformer | **4.7** | **2.3** | **106** |

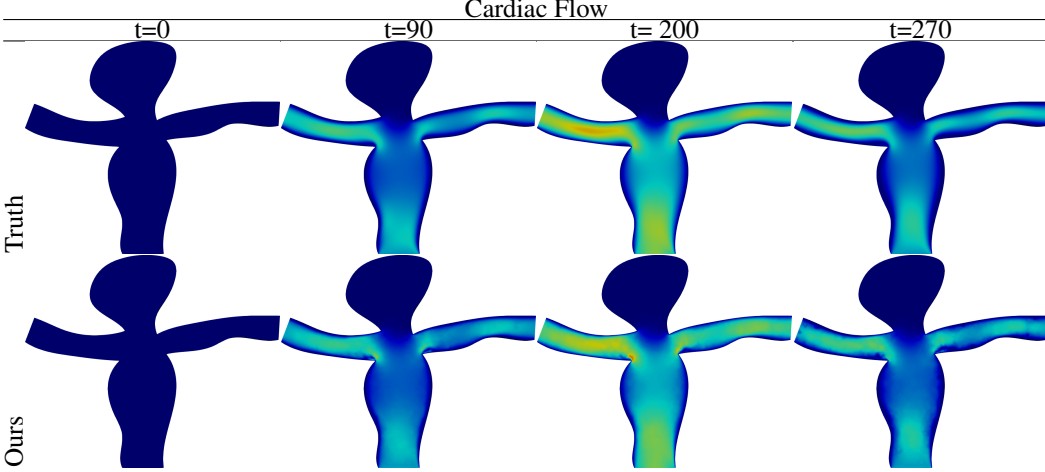

Figure 18: The velocity contours of rollouts predicted by our model versus the ground truth (CFD) for a cardiac flow.

