# OpenReview forum: "Predicting Physics in Mesh-reduced Space with Temporal Attention"
_ICLR.cc/2022/Conference — ICLR 2022 Poster_

### Official Review · Reviewer_X3ET · 2021-10-31

**Correctness:** 3
**Technical Novelty And Significance:** 2
**Empirical Novelty And Significance:** 3
**Recommendation:** 6
**Confidence:** 4

**Main Review:**

The use of a transformer architecture with attention in the context of physical simulations is interesting, and not widely studied (as far as I know). As such I was curious to see how well it works. The training methodology leaves a few open questions: is the network actually trained with sequences of different length than used for the evaluations later on? The autoregressive training described in 3.3 sounds like the network is trained on "full sequences". However, what would be important to demonstrate is that the models are trained for sequences of length n, and then evaluated for much longer sequences, e.g. 2n, 3n or higher for input parameters not seen at training time. If the training was successful, these should still give reasonable predictions.

The paper then compares a the proposed transformer with GRUs and LSTMs and comes to the conclusion that the transformer performs best across all scenarios. Unfortunately, my main worries stem from these simulation scenarios.

Interestingly, both LSTM and GRU are on-par for the medical example. This example seems quite artificial: it's basically the cylinder wake flow case, but repeated with a vessel like geometry. That seems to make it even simpler than the original wake flow case. It would have been more interesting to model something like a simple cardiac cycle.

While the summary of the different simulation cases makes sense in the main paper, the details provided in the appendix are insufficient. Neither governing equations, nor variables are properly explained. As thus, I find it very difficult to estimate whether the simulations are meaningful, which parameters were varied for the training data sets, or which cases were used for the test evaluations.

My guess is that the compressible "sonic" case is the most interesting one, but it's content remains unclear, as does the "T_,0" parameter that seems to be varied for the models. The two evolutions shown are very similar, though, and this case seems to consist of only 40 evaluation steps. That is probably too short to really leverage the attention architectures.

Despite this, the analysis of the latent space content and attention weights provide some interesting insights, and it's good to see that the dominant CFD frequency is present in the attention data.

The appendix gives additional details of the network architectures used. Unfortunately, model sizes (in terms of trainable parameters) are missing.


**Summary Of The Paper:**

This paper proposes to use auto-regressive sequence models with attention mechanisms to predict the evolution of physical simulations. Mesh-based discretizations are targeted, and hence the employed networks take the form of GNNs. The paper focuses on encode-process-decode structures, and a transformer is proposed as main method to predict future states in the latent space. This approach is evaluated with three CFD settings: an unsteady wake flow, a sonic backwards facing step, and a medical flow scenario. The results are additionally analyzed in terms of latent space and attention weight content.


**Summary Of The Review:**

Overall, I think the paper targets a very interesting direction, and I can encourage the authors to continue their work in this direction. However, in the current form I find it difficult to argue for accepting the paper straight away. The training and simulation setups leave too many open questions to properly evaluate whether the attention approach actually provides benefits or not.

Post rebuttal: the authors have answered my questions, and added two interesting simulation results in the appendix. So I've adjusted my score to a more positive one, and (seeing the other two positive reviews) I'd be fine with acceptance.

---

> ### Author Response · Authors · 2021-11-17
> **Response to Reviewer X3ET**
>
> >The compressible "sonic" case is the most interesting one, but it's content remains unclear, as does the "T_,0" parameter that seems to be varied for the models. The two evolutions shown are very similar, though, and this case seems to consist of only 40 evaluation steps. That is probably too short to really leverage the attention architectures.
>
> **Response**: We apologize for the lack of details of this case, and we add more information on the experimental setup in the revised manuscript. Yes, initial temperatures are varied for different evaluations. And different evolutions are with different fine scale features, models still need to capture subtle differences in different evolutions to achieve the small error shown in Table 1. In contrast to the cylinder flow where the periodic vortex shedding can be endless, the episode of the sonic flow ends when the moving wedge hit the wall. We agree that the error accumulation is less a problem in shorter sequences, so the advantage of our model is less pronounced in this case.
>
> >The training and simulation setups leave too many open questions to properly evaluate whether the attention approach actually provides benefits or not.
>
> **Response**: We have revised the manuscript based on the comments and marked these revisions. We hope these revisions will address your questions. Thanks for your thoughtful comments and helpful suggestions.

---

> > ### Comment · Reviewer_X3ET · 2021-11-27
> > **Post-rebuttal**
> >
> > I'd like to thank the authors for their updates. While I think there are still a few concerns with this work (also voiced in some of the other reviews), there are definitely interesting aspects here, and I'm not opposed to accepting this paper. I've raised my score to reflect this.

---

> > > ### Author Response · Authors · 2021-11-27
> > > **Response to Reviewer X3ET**
> > >
> > > We sincerely thank you so much for your support!

---

> ### Author Response · Authors · 2021-11-17
> **Response to Reviewer X3ET**
>
> Thank you for the summary of our work and constructive comments. The review is positive about the transformer architecture with attention and the analysis of the latent space and attention weights. The review also poses several questions, which we addressed in the following.
>
> >Is the network actually trained with sequences of different length than used for the evaluations later on?
>
> **Response**: Thanks for your good suggestion. We have performed such an experiment where a model trained on a sequence of length n on CylinderFlow is rolled out for 2n timesteps at test time. We observe that our model can still maintain high predictive accuracy in this case. The experiment is included in the appendix (A.11) of the updated manuscript.
>
> >Both LSTM and GRU are on-par for the medical example.
>
> **Response**: LSTM and GRU are model variants of our method, i.e. they also benefit from our contributions on mesh-reduced encoding, and are trained by passing gradients through the entire sequence. While overall, we found that the transformer variant can best make use of history information, there are cases (such as vascular flow) where LSTM/GRU variants show similar performance, and might be good choices when simplicity or faster inference times are a priority.
>
> >This example (the medical example) seems quite artificial: it's basically the cylinder wake flow case, but repeated with a vessel like geometry. That seems to make it even simpler than the original wake flow case. It would have been more interesting to model something like a simple cardiac cycle.
>
> **Response**: We agree with the reviewer that it is very interesting to model the pulsatile features in cardiac cycles, which is more realistic for cardiovascular systems. Therefore, we conducted additional experiments of pulsatile cardiac flows with varying viscosity in multiple cardiac cycles, and our models show accuracy predictions on this setups as well. We have included this additional experiment in A.13 in the revised manuscript.
>
> However, we'd like to point out the difference between case 1 (cylinder flow) and case 3 (vascular flow): case 1 is purely an external flow without wall boundaries, while the case 3 is a combination of an external flow and internal wall-bounded flow with irregular shapes and varying meshes, which breaks the symmetric vortex shedding pattern and complicates the dynamics of the flow. Hence, case 3 is actually more challenging than case 1, and the challenge can be reflected from the deteriorated prediction performance of the MeshGraphNet, which cannot accurately capture the general flow pattern (see Fig.4). Even without the pulsating cardiac flow, the problematic effect of error accumulation in next-step prediction models become more serious: in the MeshGraphNets baseline, the left inflow slowly diminishes during rollout. This is in addition to phase/frequency mismatch of the vortex shedding, which can also be observed in case 1. As we know from fluid dynamics, the instability introduced by shape is usually more challenging to predict than the unsteadiness forced by the external impulse, so we think our experiment in case 3 is still meaningful. Moreover, varying the cylinder's geometry in the middle is commonly used to model the tumor growth and cerebral infarction in hemodynamics, which has practical implications.
> >The details provided in the appendix are insufficient. Neither governing equations, nor variables are properly explained. ...  model sizes (in terms of trainable parameters) are missing.
>
> **Response**: Thank you very much for the suggestion. We have added more details on the experimental setup in the appendix A.4 and A.5, and marked these updates in the revised manuscript.

---

### Official Review · Reviewer_UQq7 · 2021-10-31

**Correctness:** 4
**Technical Novelty And Significance:** 3
**Empirical Novelty And Significance:** 3
**Recommendation:** 6
**Confidence:** 4

**Main Review:**

Strengths:
1.	The proposed model which uses GNNs as auto-encoders for the graphs and then uses transformers as dynamic models seems to be innovative.
2.	The empirical results on the three scenarios tested also support the better performance of the newly proposed model compared to MeshGraphNet, especially in the first and the third scenarios.

Weaknesses:
1.	The main critic (or confusion) I have for this model is its ability to capture complicated dynamics beyond oscillations. The transformer architecture makes this model good at handling oscillations, as earlier states can be easily copied to the current states through the attention mechanism. But what about scenarios beyond oscillations? In fact, just as the paper itself shows in the second scenario (Sonic flow), where the oscillations seem to be less happening, the difference between this algorithm seems to be less compared the MeshGraphNet (or even worse compared to it). The initial higher error also indicates that the transformer model actually fails to capture the actual dynamics. I think the authors need to show more scenarios not mainly about oscillations and whether this method still outperform the other one.
2.	More results on how the performance of the GNNs for compressing the graph representations into hidden states are needed. The authors mention that the initial higher errors might be due to the loss of the states from the auto-encoders, can the auto-encoders be modified to make this loss lower? What are some key hyperparameters on this structure that would influence the performance?
3.	Can authors also provide some failure examples of this model to help better understand its strength and weakness?
4.	This might be because I am not familiar about this specific field, but what is the main algorithm difference between this model and MeshGraphNet? Was the GNN autoencoder model also used there? Or is that newly proposed in this model?


**Summary Of The Paper:**

This work proposes a new algorithm combining graph-neural-network (GNN) and auto-regressive sequence models for physics prediction problems. The authors first use GNNs to compress the physical graphs, then use transformers to predict the next steps of the compressed representations, and finally use GNNs to recover the graph representations from the predicted representations. Through empirical studies, the authors show that this method outperform the previous SOTA model (MeshGraphNet) significantly, especially in the long-rollout prediction scenarios. The authors further analyze the models to understand the success and find that the ability of the transformer model to replay the earlier sequences seems to be the critical for the better performance especially in the scenarios with oscillations.

**Summary Of The Review:**

This work proposes a new model for prediction physics dynamics. This new model uses graph neural networks to compress the physic graphs into lower-dimensional representations and then uses transformer models to predict the dynamics. Through empirical studies on three physical scenarios, the authors show the advantage of this method compared to the previous SOTA. However, more results need to be presented to show the power of this method in complicated physical scenarios beyond oscillations, where the transformer models indeed gain advantage by their nature.

---

> ### Author Response · Authors · 2021-11-17
> **Response to Reviewer UQq7:**
>
> Thank you for the summary of our work and constructive suggestions. The review is positive about the architecture of the proposed model (GNNs as auto-encoders and transformers as dynamic models) and the significant reduction of predictive errors. The review also raises questions about the model's abilities. We respond to these questions as follows.
>
> >The main critic (or confusion) I have for this model is its ability to capture complicated dynamics beyond oscillations. ... But what about scenarios beyond oscillations?
>
> **Response**: We see several advantages of temporal attention and training on whole sequences. One of them is being able to use history for making long-term stable predictions; being able to capture oscillations accurately is one result of this. We studied this example in detail, as phase/frequency preservation is something many next-step methods particularly struggle with.
>
> But temporal attention is also helpful for other cases-- e.g. preventing drift and fading. In fig 4 (right), we notice that in MeshGraphNets the left inflow slowly diminishes during rollout, while our model can preserve the inflow state accurately.
> And more broadly, passing gradients through the entire trajectory helps learning models that are more robust to error accumulation, whereas next-step models never 'observe' accumulated error in training.
>
> To further address your concern and show if the proposed model is capable to capture the dynamics especially for non-oscillatory behavior, we tested our model on an additional prediction task of a non-oscillatory flow governed by the convection-diffusion equation. We show that our model outperforms MeshGraphNets in this domain as well. The details can be found in the newly added section A.12 in the appendix of the revised manuscript.
>
> >The initial higher error also indicates that the transformer model actually fails to capture the actual dynamics. I think the authors need to show more scenarios not mainly about oscillations and whether this method still outperform the other one.
>
> **Response**: In the second scenario, our model actually captures the underlying dynamics reasonably well. Note that the error of our model is at the level of $10^{-3}$, which is already very small. Our model shows better performance in predicting $u$ and $v$ than MeshGraphNets though its performance on $p$ and $T$ is slightly worse than MeshGraphNets. MeshGraphNets can achieve lower error at beginning of the trajectory, as it operates directly on the original mesh. But because it works in this manner, it cannot store long mesh history due to memory limitations and thus cannot capture long-term dependencies. As shown in Table 5, the recover error of the auto-encoder for Sonic flow is $1.11\times10^{-3}$, indicating the the prediction error is actually dominated by the recover error while the transformer capture the hidden dynamics very well. Due to the short rollout length in the second scenario, error accumulation in MeshGraphNets is less catastrophic, and the advantage of our model is less pronounced in this case.
>
> Additionally, as mentioned above, we have conducted a new experiment to predict non-oscillatory convection-diffusion flow.
>
> >More results on how the performance of the GNNs for compressing the graph representations into hidden states are needed. Can the auto-encoders be modified to make this loss lower? What are some key hyperparameters on this structure that would influence the performance?
>
> **Response**: We provide the recover error of the auto-encoder in A.7 (Table 5). Note that the recovery error is on a similar level to the predictive error (Table 1) for the second and third tasks. There is a tradeoff in choosing the reduced representation size: While increasing the dimensionality of the reduced graph representation can reduce the auto-encoder loss, it makes it harder to learn the temporal attention module. Hence, overall, we didn't observe significant reduction of the predictive error when increasing the size of the reduced representation and we believe 256 pivotal nodes constitute a balanced tradeoff.
>
>
> >What is the main algorithm difference between this model and MeshGraphNet? Was the GNN autoencoder model also used there? Or is that newly proposed in this model?
>
> **Reponse**: MeshGraphNets is a next-step model, that is, it directly predicts the state t+1 from the current state t using a GNN. During inference, trajectories are generated by feeding the model output back as input in a loop; this can lead to drift and accumulation of error.
>
> In contrast, our model uses a transformer to auto-regressively predict trajectories, attending to the history. We also operate in a mesh-reduced space-- MeshGraphNets does not use an autoencoder, but directly operates on the full mesh. We are the first to propose to compress the dimension of a physics graph and combine it with a sequential net.
>
> Thanks for your valuable comments and suggestions, which helped in improving the manuscript.

---

> > ### Comment · Reviewer_UQq7 · 2021-11-30
> > **response**
> >
> > Thanks for the rebuttal. I think it answers my questions well and I am happy to raise my score to 6.

---

> > > ### Author Response · Authors · 2021-11-30
> > > **Response to Reviewer UQq7**
> > >
> > > Thanks so much for your feedback and help!

---

### Official Review · Reviewer_7ipR · 2021-11-02

**Correctness:** 3
**Technical Novelty And Significance:** 2
**Empirical Novelty And Significance:** 2
**Recommendation:** 6
**Confidence:** 5

**Main Review:**

1. The paper's abstract starts with "Auto-regressive sequence models for physics prediction are often restricted to
low-dimensional systems, as memory cost increases with both spatial extents and sequence length."-- This is not true. A plethora of work in computational physics, fluid dynamics, and climate dynamics have looked at autoregressive prediction for very high-dimensional chaotic systems, and quite successfully with deep learning. The authors should do a more thorough literature review looking at these works.

2. The cases presented are not that high-dimensional. In fact, the cylinder flow can be decomposed to its first 3 POD modes which are enough to describe the dynamics of the system. Case 2 is moderately complicated, Case 3 is a more suitable test case. So, motivating this work from a perspective of high-dimensional physics is not quite fair. It is undoubtedly a good starting point, but there have been many works on much more complicated systems.

3. The authors should dig some literature to look at methods used in what can be considered state-of-the-art in this field of data-driven physics. The baselines should include some of them.

4. The best thing about the paper is the effort to diagnose what is going on in different components. I think that is a big step forward in this field. I feel this section of the paper to be very insightful and interesting.

**Summary Of The Paper:**

The paper presents a GNN-based model with temporal attention to auto-regressively predict flow velocity of 3 test cases in fluid dynamics. Prediction results and some diagnostic analysis on the components of the network have been done.

**Summary Of The Review:**

I recommend acceptance because of the diagnostic section of the paper. While there are a few concerns, I think with a bit of revision and better baselines, the paper presents a nice set of results especially with the diagnostics section

---

> ### Author Response · Authors · 2021-11-17
> **Response to Reviewer 7ipR:**
>
> We appreciate the constructive suggestions by the reviewer. The review is positive about the diagnosis of our model's components: compact representation of the full mesh and attention values corresponding to the system dynamics. The review raises a few questions about the literature study and the motivation of our work. We believe these questions can be addressed by revising our writing. We post our responses to these questions as follows and also revised our submission accordingly.
>
> >Issues in the summary of the literature on high-dimensional systems.
>
> **Response**: We thank the reviewer for pointing out this issue, and we agree that we should formulate this claim more precisely. We meant to say that ``graph-based auto-regressive sequence models'' are limited to low-dimensional systems. We generally cannot afford to directly apply multi-step sequence networks (e.g., LSTM or Transformer) to original high-dimensional physical state space represented by large-size graphs due to memory issues. On the other hand, most dimensionality reduction techniques can not easily be applied to unstructured meshes.
> Hence, current graph-based physics prediction models are one-step predictors (e.g., MeshGraphNet[1] and CP-GNet[2]). We are aware that many existing works have developed auto-regressive sequence models for high-dimensional physical space based on certain dimension reduction techniques, e.g., CNN autoencoder~([3],[4],[5],[6],[7],[8],[9],[10]), POD([11],[12],[13],[14],[15],[16],[17],[18]), or Koopman operators ([19],[20],[21]). In these works, the sequence networks are constructed in the low-dimensional latent space with either linear POD modes or nonlinear CNN autoencoders. However, these existing models cannot directly handle unstructured data (with moving or varying-size meshes), and many of them do not consider parameter variations. For example, POD cannot deal with state vectors with different lengths (e.g., varying-mesh size data in our Case 3) and is not robust for parametric cases, and CNN auto-encoder can only be applied to image-like rectangular domain with uniform grids; otherwise, rasterization/voxelization and Signed Distance Function (SDF) are required to preprocess the unstructured data, which lead to many issues and drawbacks as discussed in [23].
>
> In contrast, this work focuses on graph neural networks (GNN), which can directly handle unstructured mesh data and are very flexible for cases even with moving meshes or varying-size meshes. While with CNNs, effective and well-understood auto-encoders are readily available, this is much less true for graphs, particularly for physics prediction tasks. **One of the key contributions of this work** is our proposed Graph Mesh Reducer (GMR) and Graph Mesh Up-Sampling (GMUS) networks, which fill the gap and enable reliable dimension reduction for graph-based high-dimensional mesh data.
>
> We revised our potentially misleading statements in the abstract and introduction -- making it clear that this work focuses on GNN-based surrogate simulators. We also added more related work of autoregressive sequence models based on POD and CNNs. Since the CNN or POD-based methods cannot be directly applied to unstructured mesh data in this work (e.g., moving or varying-size meshes), we could not conduct a fair comparison.
>
>
>
> >The cases presented are not that high-dimensional... Motivating this work from a perspective of high-dimensional physics is not quite fair.
>
> **Response**: We thank the reviewer for this comment. We agree that the **intrinsic dimension** of the cylinder flow at a low Reynolds number is not very high, since the periodic vortex shedding phase can be described by its first several POD modes with fairly high accuracy. However, this work aims to develop an end-to-end GNN-based learning framework to directly handle the unstructured mesh data as graphs since the linear POD or nonlinear CNN-based dimensional reduction techniques have many limitations as mentioned above. Hence, the ``high dimension" in this work refers to that the mesh dimension (i.e., number of cells) is large, posing great challenges to GNN-based end-to-end learning. Although the intrinsic dimension of cylinder flow is relatively low, its mesh dimension is still too high to construct GNN-based sequence models. Moreover, the POD-based model reductions are known to be unstable in parametric settings -- predicting solutions with parameter variations (e.g., varying Reynolds number and boundary conditions) [22]. Considering that the cylinder flow is a classic test case in SOTA graph-based autoregressive models (MeshGraphNet[1]), we chose to start with it to demonstrate the merit and effectiveness of the proposed model. In the revised manuscript, we better clarify the motivation of dealing with high-dimension mesh data in an end-to-end manner to highlight the contribution of this work.
>
> Finally, thank you for reviewing our work. Your comments and advises are very helpful.

---

> ### Author Response · Authors · 2021-11-17
> **References**
>
>  References:
>
> [1]Tobias Pfaff, Meire Fortunato, Alvaro Sanchez-Gonzalez, and Peter W Battaglia. Learning
> mesh-based simulation with graph networks. In International Conference on Learning Representations, 2021.
>
> [2]Jiayang Xu, Aniruddhe Pradhan, and Karthik Duraisamy. Conditionally parameterized, discretization-aware neural networks for mesh-based modeling of physical systems. arXiv preprint arXiv:2109.09510, 2021.
>
> [3]Jiang-Zhou Peng, Siheng Chen, Nadine Aubry, Zhi-Hua Chen, and Wei-Tao Wu.  Time-variant prediction of flow over an airfoil using deep neural network.Physics of Fluids, 32(12):123602, 2020a.
>
> [4]Jiang-Zhou Peng, Siheng Chen, Nadine Aubry, Zhihua Chen, and Wei-Tao Wu. Unsteadyreduced-order model of flow over cylinders based on convolutional and deconvolutionalneural network structure.Physics of Fluids, 32(12):123609, 2020b.
>
> [5]Romit Maulik, Bethany Lusch, and Prasanna Balaprakash.  Reduced-order modeling ofadvection-dominated systems with recurrent neural networks and convolutional autoen-coders.Physics of Fluids, 33(3):037106, 2021.
>
> [6]Angran Li, Ruijia Chen, Amir Barati Farimani, and Yongjie Jessica Zhang. Reaction dif-fusion system prediction based on convolutional neural network.Scientific reports, 10(1):1–9, 2020.
>
> [7]Takaaki Murata, Kai Fukami, and Koji Fukagata.  Nonlinear mode decomposition withconvolutional neural networks for fluid dynamics.Journal of Fluid Mechanics, 882, 2020.
>
> [8]Kazuto Hasegawa, Kai Fukami, Takaaki Murata, and Koji Fukagata. Machine-learningbased reduced-order modeling for unsteady flows around bluff bodies of various shapes.Theoretical and Computational Fluid Dynamics, 34(4):367–383, 2020.
>
> [9]Kai Fukami, Koji Fukagata, and Kunihiko Taira. Assessment of supervised machine learning methods for fluid flows. Theoretical and Computational Fluid Dynamics, 34(4):497–519,2020.
>
> [10]Masaki Morimoto, Kai Fukami, Kai Zhang, Aditya G Nair, and Koji Fukagata. Convolutional neural networks for fluid flow analysis: toward effective metamodeling and lowdimensionalization. arXiv preprint arXiv:2101.02535, 2021.
>
> [11]Pantelis R Vlachas, Wonmin Byeon, Zhong Y Wan, Themistoklis P Sapsis, and Petros oumoutsakos. Data-driven forecasting of high-dimensional chaotic systems with long short-term memory networks. Proceedings of the Royal Society A: Mathematical, Physical and Engineering Sciences, 474(2213):20170844, 2018.
>
> [12]Xuping Xie, Guannan Zhang, and Clayton G Webster. Non-intrusive inference reduced order model for fluids using deep multistep neural network. Mathematics, 7(8):757, 2019.
>
> [13]Ashesh Chattopadhyay, Pedram Hassanzadeh, and Devika Subramanian. Data-driven predictions of a multiscale lorenz 96 chaotic system using machine-learning methods: reservoir computing, artificial neural network, and long short-term memory network. Nonlinear Processes in Geophysics, 27(3):373–389, 2020.
>
> [14]Suraj Pawar, SM Rahman, H Vaddireddy, Omer San, Adil Rasheed, and Prakash Vedula. A deep learning enabler for nonintrusive reduced order modeling of fluid flows. Physics of Fluids, 31(8):085101, 2019.
>
> [15]Pierre Jacquier, Azzedine Abdedou, Vincent Delmas, and Azzeddine Soulaïmani. Nonintrusive reduced-order modeling using uncertainty-aware deep neural networks and proper orthogonal decomposition: Application to flood modeling. Journal of Computational Physics, 424:109854, 2021.
>
> [16]Hamidreza Eivazi, Hadi Veisi, Mohammad Hossein Naderi, and Vahid Esfahanian. Deep neural networks for nonlinear model order reduction of unsteady flows. Physics of Fluids,32(10):105104, 2020.
>
> [17]Pin Wu, Junwu Sun, Xuting Chang, Wenjie Zhang, Rossella Arcucci, Yike Guo, and Christopher C Pain. Data-driven reduced order model with temporal convolutional neural network. Computer Methods in Applied Mechanics and Engineering, 360:112766, 2020.
>
> [18]Stefania Fresca and Andrea Manzoni. Pod-dl-rom: enhancing deep learning-based reduced order models for nonlinear parametrized pdes by proper orthogonal decomposition. 2022.
>
> [19]Bethany Lusch, J Nathan Kutz, and Steven L Brunton. Deep learning for universal linear embeddings of nonlinear dynamics. Nature communications, 9(1):1–10,2018.
>
> [20]Hamidreza Eivazi, Luca Guastoni, Philipp Schlatter, Hossein Azizpour, and Ricardo Vinuesa.Recurrent neural networks and koopman-based frameworks for temporal predictions in a low-order model of turbulence. International Journal of Heat and Fluid Flow, 90:108816,2021.
>
> [21]Nicholas Geneva and Nicholas Zabaras. Transformers for modeling physical systems. arXiv preprint arXiv:2010.03957, 2020.
>
> [22]Kyle Michael Washabaugh. Faster Fidelity for Better Design: A Scalable Model Order Reduction Framework for Steady Aerodynamic Design Applications. PhD thesis, Stanford University, 2016.
>
> [23]Han Gao, Luning Sun, and Jian-Xun Wang. Phygeonet: physics-informed geometry-adaptiveconvolutional neural networks for solving parameterized steady-state pdes on irregular domain, 2021.

---

### Official Review · Reviewer_PRtV · 2021-11-08

**Correctness:** 4
**Technical Novelty And Significance:** 3
**Empirical Novelty And Significance:** 3
**Recommendation:** 8
**Confidence:** 4

**Main Review:**

This paper tackles the problem of simulating physics using graph neural networks (GNNs). GNNs have recently shown to be extremely efficient and fast alternatives to physical simulators. The current state-of-the-art of physical simulation with neural networks, is MeshGraphNets, which uses a GNN archictecture to learn one-time-step updates. These updates are integrated over time to produce arbitrary length simulations.

However, the feedforward nature of MeshGraphNets brings two issues to the fore.
* Long-term stability: Since MeshGraphNets do not incorporate any feedback mechanism, they operate in an open-loop system, often resulting in eventual divergence of predicted velocities/accelerations.
* Physical implausibility: The training recipes of MeshGraphNets requires adding noise to the state vectors (e.g., perturbations to positions velocities etc.) without which the approach overfits to a set of reference trajectories. Adding noise to these physical quantities makes the learning setup physically implausible (i.e., the function we learn does not map physically accurate initial states to physically accurate subsequent states.

This paper addresses both these issues by incorporating a self-attention mechanism between the graph encoder and decoder layers.

Another key novel aspect of the paper is in the reduction step, where an input graph is reduced to a much smaller, but representative ‘summary graph’. Learning dynamics over the summary graph, in principle, is sufficient to represent dynamics over the full mesh.

The experiments and results presented in this paper provide compelling evidence to both claims, and as such I would recommend this paper for acceptance. However, there are a few minor discussion points / analyses I believe will further strengthen the paper.

Is there any insight on how succinct the learned representation might be? For instance, in the case of rigid bodies, the input meshes may be subsumed into a single node (the center-of-mass) and yet represent the full system dynamics.

Does the sampling strategy of the summary nodes have a significant impact on the resulting learned physical model? For objects that are inherently non-uniform density, the graph network might benefit from sampling in proportion to the object density as opposed to uniformly. Any thoughts/insights on this?

**Summary Of The Paper:**

This paper presents a graph and attention based architecture to model the dynamics of complex physical systems. Their experiments indicate a significant (in cases, upto two order of magnitude) outperformance of a state-of-the-art graph network baseline.

**Summary Of The Review:**

This paper is very well-written, tackles an important problem, and overcomes two shortcomings in existing approaches.

---

> ### Author Response · Authors · 2021-11-17
> **Response to Reviewer PRtV:**
>
> Thank you so much for the summary of our work and insightful suggestions. The review is positive about the architecture of our model, the significant reduction of predictive error, and the compact representation of the full mesh.  Here are our responses to your questions.
>
> >Is there any insight on how succinct the learned representation might be?
>
> **Response**: We have an error analysis of the encoder-decoder structure. With our current model settings on three tasks, the compression ratio of the encoder is about 25 ~ 37 across three datasets, while the recovery error (RMSE) from the decoder is only 0.001 ~ 0.014 (Table 5 in the appendix).
>
> >Does the sampling strategy of the summary nodes have a significant impact on the resulting learned physical model? ... the graph network might benefit from sampling in proportion to the object density as opposed to uniformly.
>
> **Response**: Yes, you are right. We find that it's important to allocate the pivotal nodes in regions where nontrivial dynamics and steep gradients are expected. This is one of the advantages of using irregular simulation meshes that we want to also exploit in our model.
>
> Our strategy is generally to sample uniformly **from the fine simulation mesh**. We note that this doesn't result in a uniform sampling in space, but a sampling proportional to the node density of the simulation mesh (see Fig. 10-12 in the appendix). We believe this is already a good heuristic, as the simulation mesh already refines its resolution in  regions of complex dynamics (e.g. around the cylinder), and by proportional sampling we can carry this over. We have made this clearer in the revised version of the paper. Thanks for your suggestions.
>
> We agree that it would be interesting to explore this concept further; e.g. could we learn an optimal assignment of pivotal nodes which optimizes prediction performance? It would be exciting to study these questions in future work.
>
> We want to thank the reviewer for carefully evaluating our work and for kind words about it! Please see the revised version of the manuscript in the pdf file. All significant changes are in blue.

---

### Public Comment · ~Jiayang_Xu1 · 2021-11-10
**Happy to find this nice architecture**

So happy to find this during my literature review on deep learning for computational physics. Actually it covers all mainstream techniques that I am aware of in this field, dimensionality reduction, GNN, autoregression, attention mechanism, and combined them very naturally. Ultimately, I think this aligns with where the future work will go -- a completely graph-based model would not be so efficient per time-step (a large portion of acceleration come from larger time step sizes compared to a solver), and a classic autoencoder based autoregressive model often lacks portability. This combination has the potential to combine the advantages of both, and actually it has been demonstrated to outperform a strong baseline, Meshgraphnets. In such a combination of novel methods it can be hard to distinguish the contribution of each component, but not the case here -- the diagnose part is great can clearly explains how the graph reduction operates in a different manner than the POD, and how attention behaves. The results are much convincing with them, but it is still quite impressive that a autoregressive model can deal with changing meshes and advection problems. Is this because that the spatial structure is preserved to some degree through the graph reduction-upsampling? This method looks elegant, and seems very easy to use compared to many existing convolutional autoencoders that need a lot artificial treatments at the boundaries. Still, it surprises me that no visible error showed up near the boundaries even with a change in number of edges per node. I wonder if any additional treatments are necessary at all for the boundaries, e.g., node-type labels, feature normalizations? Or are the BC fixed/read-in? I Look forward to learn more from your reply, and good luck!

---

> ### Author Response · Authors · 2021-11-17
> **Thank you so much for your detailed comments and evaluations.**
>
> First of all, thank you so much for your interest in our paper. We did not use additional treatments for BC, and the performance seems well. However, what you mentioned is definitely compatible with our framework (GMR and GMUS), and we are excited about exploring these new potential treatments. We also have uploaded a new version of the paper. Finally, we appreciate your positive comments on our work.

---

### Author Response · Authors · 2021-11-25
**Revised manuscript uploaded and response to reviewers' comments posted**

Dear reviewers,

Thanks for your constructive comments and suggestions, which are helpful for improving our manuscript. We have posted the response and revised version of our paper. There are many extra new experiment results added to the paper. We also revised some clarifications and added more experiment details accordingly. We are looking forward to your further feedback. Please do feel free to let us know if you have any thoughts.

Thanks for your efforts.

Best regards,

---

### Public Comment · ~Robert_Brockhoff1 · 2022-05-03
**Questions regarding boundary conditions, training time**

First, congrats for the acceptance and the nice paper. The paper was a nice read and pretty inspiring. However, I just want to come back to a previous question and ask for the boundary conditions, specifically for the obstacle flow. Do you use No-Slip boundary conditions on the walls? Or was the domain unbounded in the y-direction? Secondly, could you report training times for your models, if available? I am quite curious, about how long you trained the architecture. It seems, that even on the reduced latent space, the maximum sequence length can become quite long if one considers long roll-outs.

Again: very nice paper, I am looking forward to furthering extensions of the idea.

Best regards

---

> ### Public Comment · ~Han_Gao3 · 2022-05-06
> **Thanks for your interest**
>
> Thanks for your interest.
> For the numerical simulations to generate the data, yes, it is no-slip bc. For cylinder flow, yes, the domain is unbounded in Y-direction. However, in this work, we do not do any specific treating on deep neural networks and let the model learn the velocity close to the boundary. It did well in our 3 cases. But you are right, and we can explicitly specify the boundary condition in the learning. See these two papers:
> Learning mesh-based simulation with graph networks by Pfaff
> Conditionally parameterized, discretization-aware neural networks for mesh-based modeling of physical systems by Xu. Our framework can incorporate such boundary treatment.
>
> As for the training cost, it varies from case to case. It can take several days for the flow with more unstable, small-frequency physics to train the model. However, it can take only several hours for the flow with more large-frequency physics. For the same case, you are right; the longer rollout you train, the more cost of training is.

---

### Decision · Program_Chairs · 2022-01-20

**Decision:**

Accept (Poster)

**Comment:**

This paper proposes a model to predict the spatiotemporal dynamics of physical simulations on irregular meshes. The observations are modeled as a sequence of graph representations, each graph corresponding to a snapshot of the observation sequence at time t. This model uses two components, a graph encoder-decoder to compress the observations and an autoregressive transformer to model the dynamics. The two components are trained sequentially. At inference time, given an initial hidden state infered from the data and some additional conditional information, a sequence of states is predicted in an auto-regressive manner in the hidden space, each state of the sequence is then decoded to produce a prediction in the original observation space. The originality of the model lies in the encoder-decoder graph and in the use of a transformer for the prediction. Tests are performed on three fluid dynamics simulation data sets.

All reviewers pointed out some original contributions in the proposed method, in particular the use of transformers in the learned hidden space. In the rebuttal, the authors provided substantial additional results and further details and explanations. Their responses led two reviewers to increase their scores. All reviewers ultimately agree that the paper presents interesting results and conclusions.